# c-JUN is a barrier in hESC to cardiomyocyte transition

Hui Zhong[1,3,*], Ran Zhang[1,2,4,*], Guihuan Li[2,*], Ping Huang[9], Yudan Zhang[4], Jieying Zhu[3], Junqi Kuang[5], Andrew P Hutchins[6], Dajiang Qin[2,4,8], Ping Zhu[1,7], Duanqing Pei[5], Dongwei Li[1,2]

**Loss of c-JUN leads to early mouse embryonic death, possibly because of a failure to develop a normal cardiac system. How c-JUN regulates human cardiomyocyte cell fate remains unknown. Here, we used the in vitro differentiation of human pluripotent stem cells into cardiomyocytes to study the role of c-JUN. Surprisingly, the knockout of c-JUN improved cardiomyocyte generation, as determined by the number of TNNT2+ cells. ATAC-seq data showed that the c-JUN defect led to increased chromatin accessibility on critical regulatory elements related to cardiomyocyte development. ChIP-seq data showed that the knockout c-JUN increased RBBP5 and SETD1B expression, leading to improved H3K4me3 deposition on key genes that regulate cardiogenesis. The c-JUN KO phenotype could be copied using the histone demethylase inhibitor CPI-455, which also up-regulated H3K4me3 levels and increased cardiomyocyte generation. Single-cell RNA-seq data defined three cell branches, and knockout c-JUN activated more regulons that are related to cardiogenesis. In summary, our data demonstrated that c-JUN could regulate cardiomyocyte cell fate by modulating H3K4me3 modification and chromatin accessibility and shed light on how c-JUN regulates heart development in humans.**

## Introduction

Numerous discoveries have led to an understanding of the regulatory network that governs cardiomyocyte cell fate decisions (1, 2, 3, 4, 5). However, the process in humans is obscured during early embryogenesis and is not amenable to genetic manipulations to uncover the regulatory mechanisms behind cardiomyocyte generation. Human embryonic stem cells (hESC) and human induced pluripotent stem cells can be differentiated into cardiomyocytes in vitro, thus providing a model system for studying the gene network regulating cardiomyocyte fate decisions and cardiomyocyte regeneration (6).

During cardiomyocyte formation, there is a complex transcriptional regulatory system involving multiple transcription factors (TFs) that control the progression through the hESC-cardiac mesoderm-cardiomyocyte stages. Particularly critical are key TFs, such as the *GATA* family/*MESP1*/*ISLET1*(*ISL1*) families of TFs. *GATA* family TFs such as *GATA4/6* play essential and evolutionary conserved roles in heart development by regulating early mesoderm formation and the mesoderm-to-cardiac fate transition (7). Activating *MESP1* in hESC promotes the epithelial-mesenchymal transition to generate mesoderm progenitors and ultimately cardiovascular cells (8, 9). *ISL1* is expressed at the dorsal pharyngeal mesoderm and helps specify the second heart field (10, 11). When combined, overexpression of the key TFs *GATA4*/*MEF2C*/*TBX5* and *MESP1*/*BAF60C* could reprogram cardiac fibroblasts toward an induced cardiomyocyte fate by remodeling chromatin structure (12, 13, 14). These data demonstrate those TFs play an important role in cardiac mesoderm and cardiomyocyte generation. However, a comprehensive understanding of the regulatory network in cardiomyocyte generation and maturation requires further study.

*c-JUN*/*AP-1* TFs are essential for mouse embryonic development. Knockout of *Jun* (*c-Jun*) in mice leads to embryonic lethality around E13.5 (15, 16). Suggesting a fundamental role for *c-JUN* in embryonic development. In the adult mouse, tissue-specific mutation of *c-JUN* leads to heart disease such as dilated cardiomyopathy (17). Recent studies have implicated the *c-JUN*/*AP-1* family of TFs in heart development, as repressing *c-JUN*/*AP-1* TFs reduced cardiomyocyte regeneration in zebrafish (18). Interestingly, (19) reported that *c-JUN*/*AP-1* binding motifs are needed for injury responses and cardiac regeneration in zebrafish and killifish, but mammals have lost (or never gained) this regenerative capability (19). During the differentiation of hESC to endoderm, *c-JUN* occupies hESC enhancers in partnership with pluripotent TFs and impedes the

[1]Guangdong Cardiovascular Institute, Guangdong Provincial People's Hospital (Guangdong Academy of Medical Sciences), Southern Medical University, Guangzhou, China   [2]Key Laboratory of Biological Targeting Diagnosis, Therapy and Rehabilitation of Guangdong Higher Education Institutes, The Fifth Affiliated Hospital of Guangzhou Medical University, Guangzhou, China   [3]CAS Key Laboratory of Regenerative Biology, Center for Cell Lineage and Development, Guangzhou Institutes of Biomedicine and Health, Chinese Academy of Sciences, Guangzhou, China   [4]Bioland Laboratory Guangzhou Regenerative Medicine and Health Guangdong Laboratory, Guangzhou, China   [5]Laboratory of Cell Fate Control, School of Life Sciences, Westlake University, Hangzhou, China   [6]Department of Systems Biology, School of Life Sciences, Southern University of Science and Technology, Shenzhen, China   [7]Guangdong Provincial Key Laboratory of Pathogenesis, Targeted Prevention and Treatment of Heart Disease and Guangzhou Key Laboratory of Pathogenesis, Targeted Prevention and Treatment of Heart Disease, Guangzhou, China   [8]Centre for Regenerative Medicine and Health, Hong Kong Institute of Science & Innovation, Chinese Academy of Sciences; Hong Kong, China   [9]The Second Affiliated Hospital of Guangzhou University of Chinese Medicine, Guangzhou, P.R. China

Correspondence: tanganqier@163.com; peiduanqing@westlake.edu.cn; lidongwei@gzhmu.edu.cn
*Hui Zhong, Ran Zhang, and Guihuan Li are Co-first authors

decommissioning of ESC enhancers, thus inhibiting hESC differentiation (20). However, it is unclear how *c-JUN/AP-1* family TFs regulate human cardiomyocyte development.

Here, using an established cardiomyocyte differentiation protocol (21, 22), we found that the knockout of *c-JUN* significantly improved TNNT2+ cell generation to >90% purity. Interestingly, when we studied the mechanism behind how *c-JUN* inhibited cardiomyocyte generation, we found that the knockout of *c-JUN* activated *RBBP5* and *SETD1B*, two members of the SET histone methylase complex, and elevated H3K4me3 levels. When analyzing the genome-wide H3K4me3 modification, we found that key factors that regulate cardiogenesis were more rapidly activated in response to the increased H3K4me3 and increased chromatin accessibility when *c-JUN* was knocked out. The *c-JUN* knockout effect could be phenocopied using the small molecule CPI-455, a histone lysine demethylase (*KDM5*) inhibitor, which caused increased H3K4me3 levels and accelerated the differentiation of hESC into cardiac mesoderm. Finally, using single-cell RNA-sequencing, we found 13 kinds of cell types that appeared during the hESC-to-cardiomyocyte transition and constructed three trajectories for the differentiating cells. These fates ended in cardiomyocyte, endothelial, and epicardial cells. Comparing the trajectory of hESC to cardiomyocytes, we found more key regulators related to cardiogenesis in the *c-JUN* KO than WT differentiation, such as *EOMES*, *GATA6*, and *HEY*. Overall, these results indicate that *c-JUN* impairs cardiomyocyte development by modulating the deposition of H3K4me3 on key genes.

# Results

## Differentiation of hESC into cardiomyocytes

hESC can be differentiated in vitro into cardiomyocytes (21, 22, 23, 24). Using a high-efficiency cardiomyocyte differentiation protocol (21), by day 15, ~70% of cells were TNNT2+ and showed spontaneous contractions (Fig 1A and B and Video 1). During cardiomyocyte differentiation, there are three major developmental stages: pluripotent (hESC, marker gene: *POU5F1* [*OCT4*]), cardiac mesoderm (marker gene: *MESP1*), and cardiomyocyte stage (marker gene: *TNNT2*) (Fig 1A). We harvested cells at D1, 3, 5, 7, and 15, along with hESC as control, and processed them for RNA-sequencing (RNA-seq). Pluripotent genes such as *OCT4/SOX2* were quickly down-regulated after the cardiac mesoderm stages, while the cardiac mesoderm genes *TBXT* (*T*)/*MESP1*/*DKK1* were immediately up-regulated during the early stages. Mature Cardiomyocyte marker genes, such as *ISL1/NKX2-5/TNNT2/MYH6/ACTN2*, were activated during the late stage, suggesting functional cardiomyocyte formation (Fig 1C). Immunofluorescence results show D15 cells expressed α-ACTININ and TNNT2 and had sarcomere-like structures (Fig 1D). To gain mechanistic insight into the cardiomyocyte generation, we divided the time-course RNA-seq data into three stages (C1–C3). The 2506 genes in the C3 group that were down-regulated from D3 onward were related to DNA replication and cell division (Fig 1E and F). The 519 genes expressed in C2 (cardiac mesoderm) were related to somitogenesis and mesoderm formation (Fig 1E and F). Finally, the 2,586 genes in C1 activated during the late

cardiomyocyte stages were identified as involved in the regulation of heart development and sarcomere organization (Fig 1E and F). These results demonstrate an efficient cardiomyocyte generation system that is suitable for further study of the function of *c-JUN* in cardiomyocyte development.

## *c-JUN* is a barrier during cardiomyocyte differentiation

*c-JUN* is required for mouse embryonic development, especially hepatocyte formation (15, 16). The role of *c-JUN* in cardiomyocyte differentiation remains unknown. Here, we found *c-JUN* was expressed throughout the process of hESC to cardiomyocyte differentiation (Fig 2A). To test whether *c-JUN* could affect cardiomyocyte differentiation, we deleted the *c-JUN* CDS in hESC with CRISPR/Cas9, generated two *c-JUN* KO (#2 and #10) clones, and confirmed the loss of c-JUN by Western blot (Fig 2B).

We then differentiated the hESC into cardiomyocytes in vitro and surprisingly found that WT and *c-JUN* KO cells were morphologically different (Fig 2C). Interestingly, we observed that spontaneous contractions of *c-JUN* KO cardiomyocytes appeared to be increased compared with WT cells (Video 1 and Video 2). Next, we quantified the level of the cardiomyocyte marker *TNNT2* and showed that *c-JUN* KO cells have a higher proportion of TNNT2+ cells than WT, ~95% versus ~74% (Fig 2D and E). To test whether *c-JUN* activation is a barrier during hESC-to-cardiomyocyte transition, we repressed *c-JUN* N-terminal kinase (JNK) activity by small molecular SP600125; the result showed that the inhibition of JNK also enhanced cardiomyocyte generation (Fig 2F). However, the DMSO appears to impair cardiomyocyte differentiation, as the number of TNNT2+ cells in the control group was lower compared with untreated cells (Fig 2E and F). On the contrary, overexpressing *c-JUN* could inhibit normal cardiomyocyte morphology and severely inhibit TNNT2+ cell generation (Fig S1A–C). Nonetheless, reduced *c-JUN*, either through the knockout or inhibition of JNK, led to increased numbers of TNNT2+ cells. Immunofluorescence supported these observations as both TNNT2 and α-ACTININ were lower in WT cells compared with *c-JUN* KO cells, and the *c-JUN* KO cells also had clearer sarcomere-like structures (Fig 2G and H). Together, these findings suggest that *c-JUN* behaves as a barrier to cardiomyocyte generation.

## Chromatin dynamics during cardiogenesis

To understand how *c-JUN* restrains cardiomyocyte differentiation, we performed RNA-seq and ATAC-seq (25, 26) on *c-JUN* KO samples, including *c-JUN* KO hESC (D0), D1, D3, D5, D7, and D15 (Fig 3A). We clustered the open/closed loci into three groups based on the chromatin changes during differentiation (Fig 3B). The first group contains pluripotent genes such as *OCT4* that is Open in the pluripotency stage and Closed in the later stage (OC) (Fig 3A). The second group contains genes such as *MESP1*, that is closed in the pluripotency stage and cardiomyocyte stage but Transiently Open (TO) in the cardiac mesoderm (Fig 3A). Third group contains mature cardiac markers, such as *TNNT2*, that are Closed in the early stage but Open at the final cardiomyocyte stage (CO) (Fig 3A). Using these definitions, a clear pattern of chromatin dynamics appeared during the hESC-to-cardiomyocyte transition. The largest group of chromatin changes was the OC group (38437 loci), followed by the

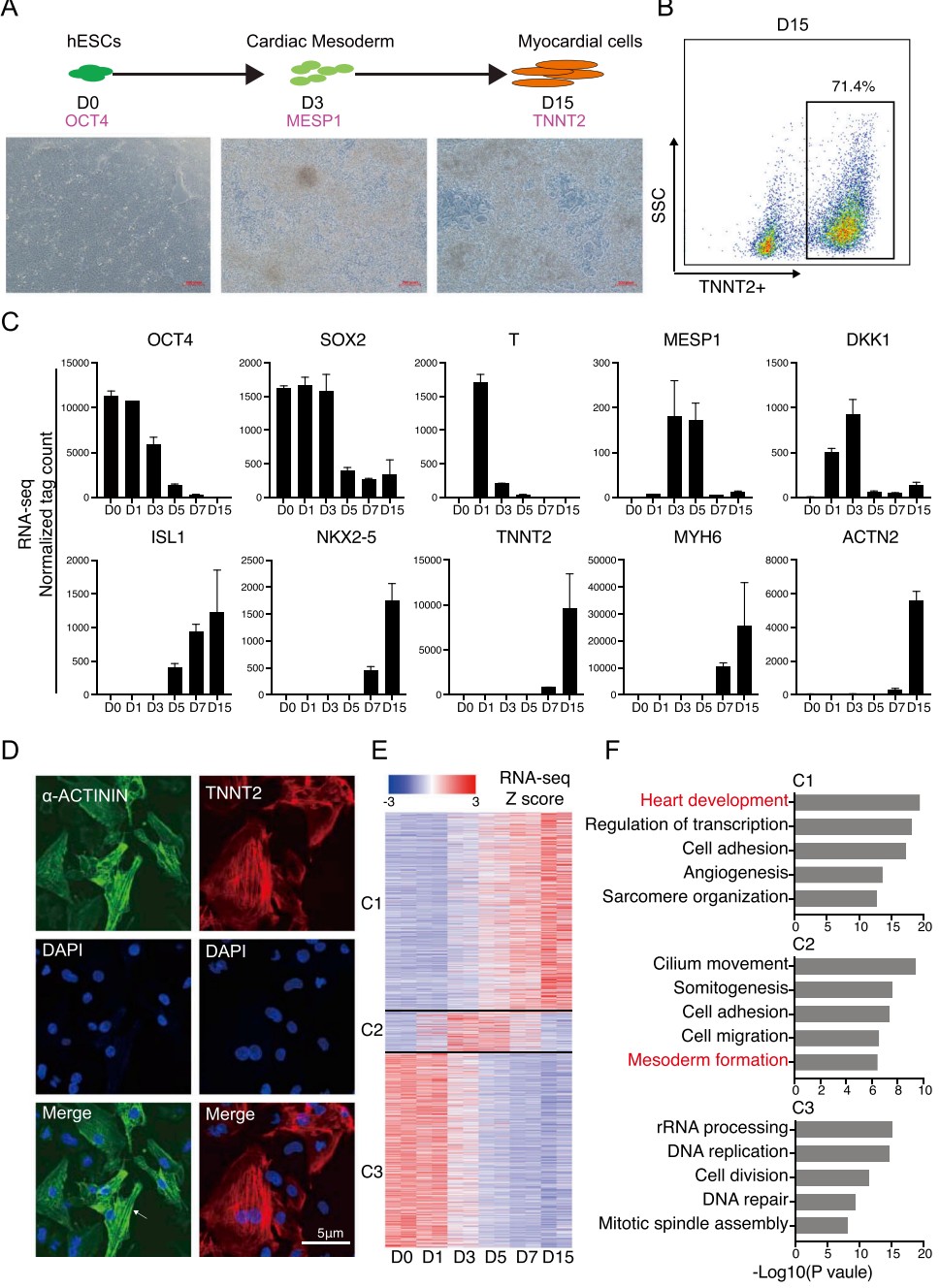

**Figure 1. Induction of human embryonic stem cells (hESC) differentiated into cardiomyocytes.**
**(**A**)** Schematic for cardiomyocyte generation from human embryonic stem cells. Below is the cell morphology at different stages; D, day. **(B)** Flow cytometry quantification of cardiomyocyte differentiation efficiency according to the GiWi method (21). **(C)** RNA-seq data showing the expression profile of representative genes during hESC-to-cardiomyocyte differentiation. Data were from two replicates and are shown as the mean. **(D)** Immunostaining showed protein expression of the cardiomyocyte marker genes TNNT2 and α-ACTININ. Scale bar, 5 μm. The white arrow point to a sarcomere. **(E)** Heatmap shows the difference expressed genes during cardiomyocyte generation. According to the expression profile, genes expressing from low to high (C1, 2586) and high in the middle stage (C2, 519) and from high to low (C3, 2506) were clustered together. **(F)** Gene ontology analysis shows the different biology functions of the genes in C1~C3 category.

CO (11618 loci), and finally the TO group (8356 loci) (Fig 3C). In agreement with our previous reports (27), the chromatin signature of the starting cells closed rapidly upon differentiation, while the target cell type chromatin structure was slower to emerge.

Chromatin accessibility is at least partially driven by the activity of specific TFs, so we searched for the DNA-binding motifs of TFs that were enriched in the OC/TO/CO categories (28). As expected, the OC group was generally dominated by key pluripotency-related factors, such as *SOX* and *OCT* TFs. Motifs representing the *GATA* family of TFs were enriched in TO and CO groups, representing the emergence of the definitive mesoderm (Fig 3D). Finally, loci in the

CO category were enriched for binding motifs for *ISL1/NKX2-5/ MEF2A/MEF2C* TFs, which are reported to play a critical function in cardiogenesis (11, 12, 14, 29, 30, 31, 32) (Fig 3D). Overall, these results suggest that the specific chromatin signature of cardiomyocytes has been established.

### *c-JUN* inhibits cardiogenesis chromatin open

Next, to study how *c-JUN* regulates cardiomyocyte differentiation, we compared the ATAC-seq data between WT and *c-JUN* KO conditions and found that the loss of *c-JUN* led to increased chromatin

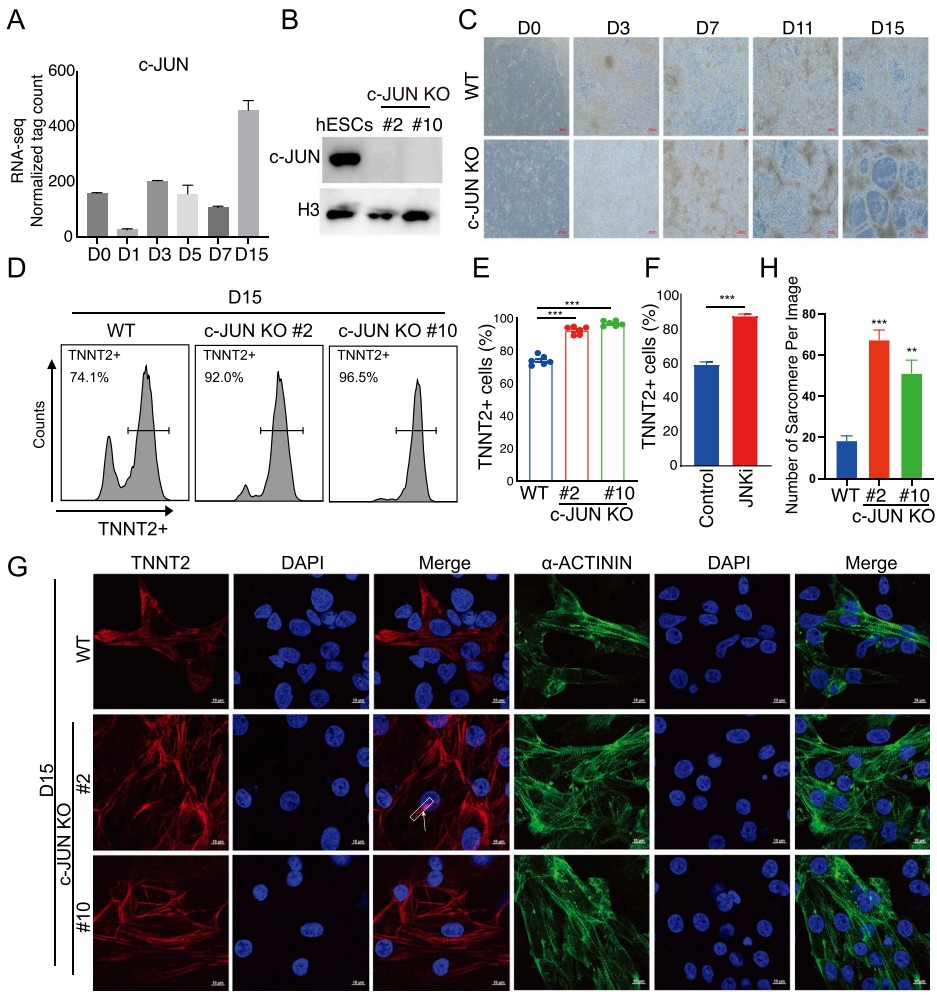

**Figure 2. c-JUN is a barrier in cardiomyocyte differentiation.**
**(A)** Bar plot shows *c-JUN* expression pattern during cardiomyocyte generation. Data were from two replicates and shown as the mean ± SEM; D, day. **(B)** Western blot for c-JUN in WT and *c-JUN* knockout hESC clones. **(C)** Morphology of cardiomyocytes differentiated from WT and *c-JUN* knockout hESC. D, day; Scale bars, 200 μm. **(D)** Flow cytometry analysis of *TNNT2* expression in cardiomyocytes differentiated from hESC. Percent of cell TNNT2+ is indicated at each condition; WT, induce cardiomyocytes from WT hESC; *c-JUN* KO, induce cardiomyocytes from *c-JUN* knockout hESC; D, day. **(D, E)** Flow cytometry quantification of differentiation efficiency from (panel (D)). Knockout of *c-JUN* enhances the number of TNNT2+ cells on D15 of the cardiomyocyte differentiation. Data were from six biological replicates in three independent experiments and are shown as the mean ± SEM. *** *P*-value < 0.001, two-way ANOVA with Sidak correction between the WT control and *c-JUN* KO groups. **(F)** Flow cytometry quantification of cardiomyocyte differentiation efficiency with or without JNK inhibitor (SP600125, 2 μm). JNKi was added to the culture medium from D0 to D15. Data were from seven biological replicates in two independent experiments and are shown as the mean ± SEM. *** *P*-value < 0.001, two-way ANOVA with Sidak correction between the control (DMSO) and JNK inhibitor groups. **(G)** Immunostaining showing knockout *c-JUN* promotes expression of cardiomyocyte marker genes TNNT2 and α-ACTININ. Scale bar, 10 μm. The white box marks a sarcomere. **(H)** Counts of the number of sarcomeres in each image. Data were from four to six replicates and are shown as the mean ± SEM. *** *P*-value < 0.001, ** *P*-value < 0.01, two-way ANOVA with Sidak correction between the WT and *c-JUN* KO groups.
Source data are available for this figure.

accessibility within the CO as early as in D3 (Fig 3A and B). The TO groups also opened earlier and were more intense (Fig 3B). To understand the impact of the loss of *c-JUN* on chromatin accessibility, we divided the CO peaks into increased accessibility and decreased accessibility groups based upon the difference between the WT and *c-JUN* KO at D15 (Fig 3E). DNA-binding motifs for key cardiogenesis TFs *TEAD/GATA* and *MEF* were enriched in the *c-JUN* KO increased accessibility CO loci (Fig 3F). Surprisingly, motifs for *AP-1* TFs were enriched in both increased and decreased CO loci (Fig 3F). This suggests the *AP-1* binding loci are more available for remodeling in the absence of *c-JUN*. Supporting an enhanced differentiation, gene ontology analysis showed that the increased accessibility of CO loci was related to cardiac muscle tissue development and cardiocyte differentiation (Fig 3G). And decreased CO loci were related to transforming growth factor signaling pathway and placenta development (Fig 3G).

Chromatin remodeling regulates and often precedes gene expression programs (27). Analysis of transcriptome dynamics by principal component analysis indicated a similar overall trajectory for the WT and KO cells (Fig 3H). However, the KO cells appeared to be accelerated, particularly around D3–D5 (Fig 3H). This agrees with the faster and more intense changes in chromatin accessibility

seen in the ATAC-seq data (Fig 3B). Differentially expressed genes (DEG) detected on D15 showed that cardiomyocyte-related genes *TNNT2/HAND1/MYH7* were significantly up-regulated in *c-JUN* KO cells, whereas other cell fate-related genes, such as *LEFTY2*, *COL6A1*, *FOXC1*, and *TET1*, were up-regulated in WT cells (Fig 3I). To identify whether those DEGs were regulated by increased or decreased accessibility CO loci, we found 51 genes up-regulated in c-JUN KO cells and also regulated by increased accessibility loci (Fig 3J), including cardiac muscle contraction-related genes *ATP1B1*, *TNNI1*, *MYH7*, and sarcomere organization-related genes *CSRP3*, *SYNPO2L*, *LDB3*, and *MYLK3* (Fig 3K). These results suggest that c-JUN inhibits cardiomyocyte differentiation by repressing chromatin opening and activating non-cardiogenesis genes.

## *c-JUN* prevents H3K4me3 modification by repressing SET complex activation

As *c-JUN* is blocking chromatin opening, we next explored if *c-JUN* was mediating this function through histone modifications. Western blot of H3K4me3 and H3K27ac in WT and *c-JUN* KO hESC showed that H3K4me3 was up-regulated in *c-JUN* KO cells (Fig 4A).

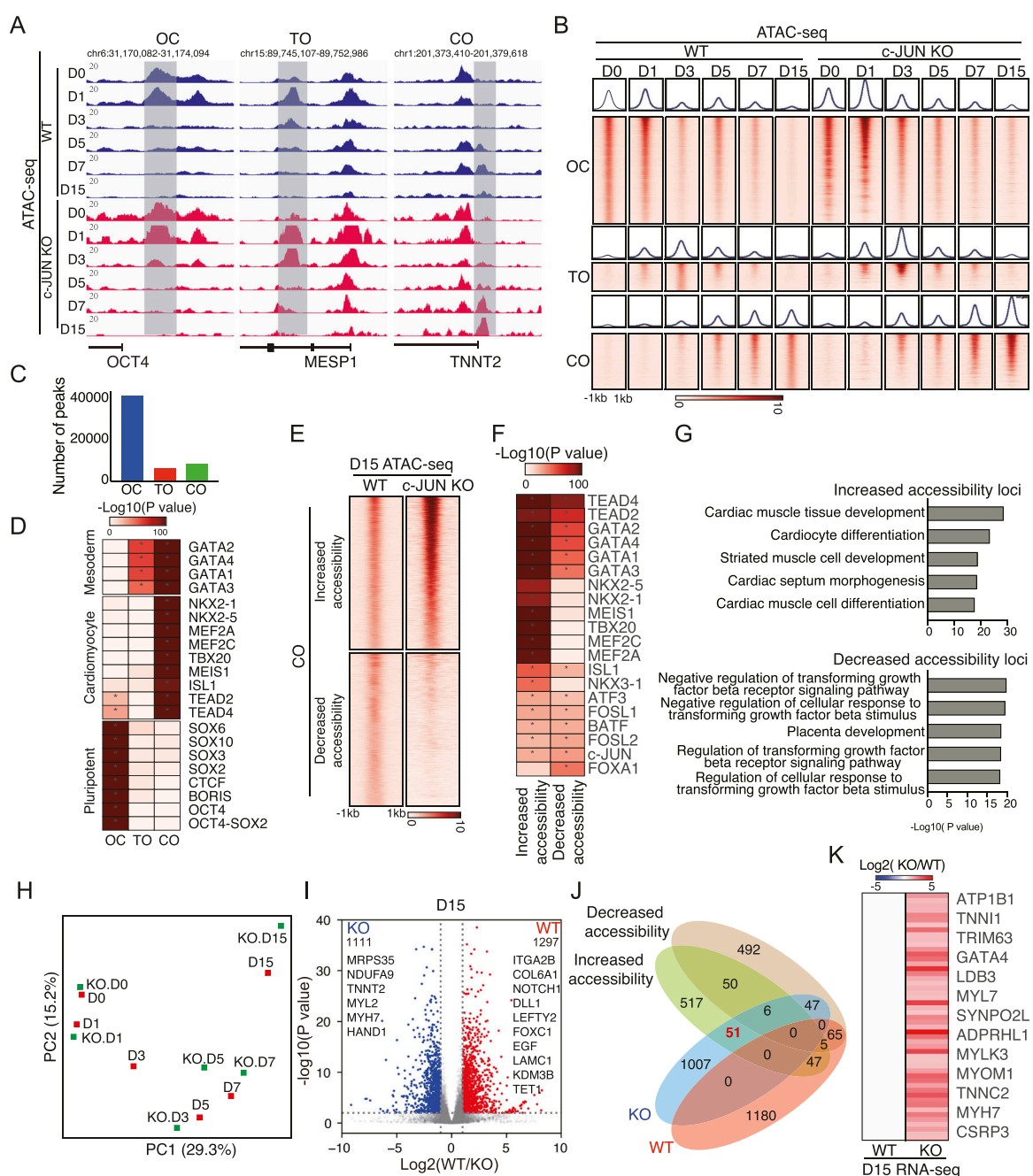

**Figure 3. *c-JUN* regulates chromatin opening during cardiomyocyte differentiation.**
**(A)** Genome view of the ATAC-seq data for Open to Close (OC), Transition Open (TO), and Close to Open (CO), represented by the pluripotent gene *OCT4*, mesoderm gene *MESP1*, and cardiomyocyte marker gene *TNNT2*. **(B)** Heatmap and pileup showing the ATAC-seq chromatin accessibility for D0 to D15 cells at OC/TO/CO loci. The heatmap and pileups are centered on the ATAC-seq peaks center. **(C)** Number of peaks in each OC/TO/CO category. **(D)** TF motifs significantly enriched at least > 1.5-fold for OC/TO/CO categories of ATAC-seq peaks. * Indicates a *P*-value < 1e-20. **(E)** Heatmap showing knockout *c-JUN* promotes half of the CO loci more accessibility (Increased accessibility) and closed half of the CO loci (Decreased accessibility). **(E, F)** TF motifs were significantly enriched at least > 1.5-fold for Increased and Decreased loci as defined in panel (E). * Indicates a *P*-value < 1e-20. **(G)** Gene ontology (GO) analysis of the significantly enriched biological process terms related to Increased accessibility and Decreased accessibility loci. **(H)** Principal component analysis of the time courses of gene expression of cardiomyocytes generated from WT and *c-JUN* knockout hESC. **(I)** Volcano plot showing the differentially expressed genes on Day 15 (D15) cardiomyocytes generated from WT and *c-JUN* KO hESC. **(I, J)** Venn diagram showing the overlap of genes regulated by Increased and Decreased loci with differentially expressed genes (defined in (I)) in WT and *c-JUN* KO cells. **(J, K)** Heatmap showing the fold-change of gene expression of the 51 genes defined in panel (J).

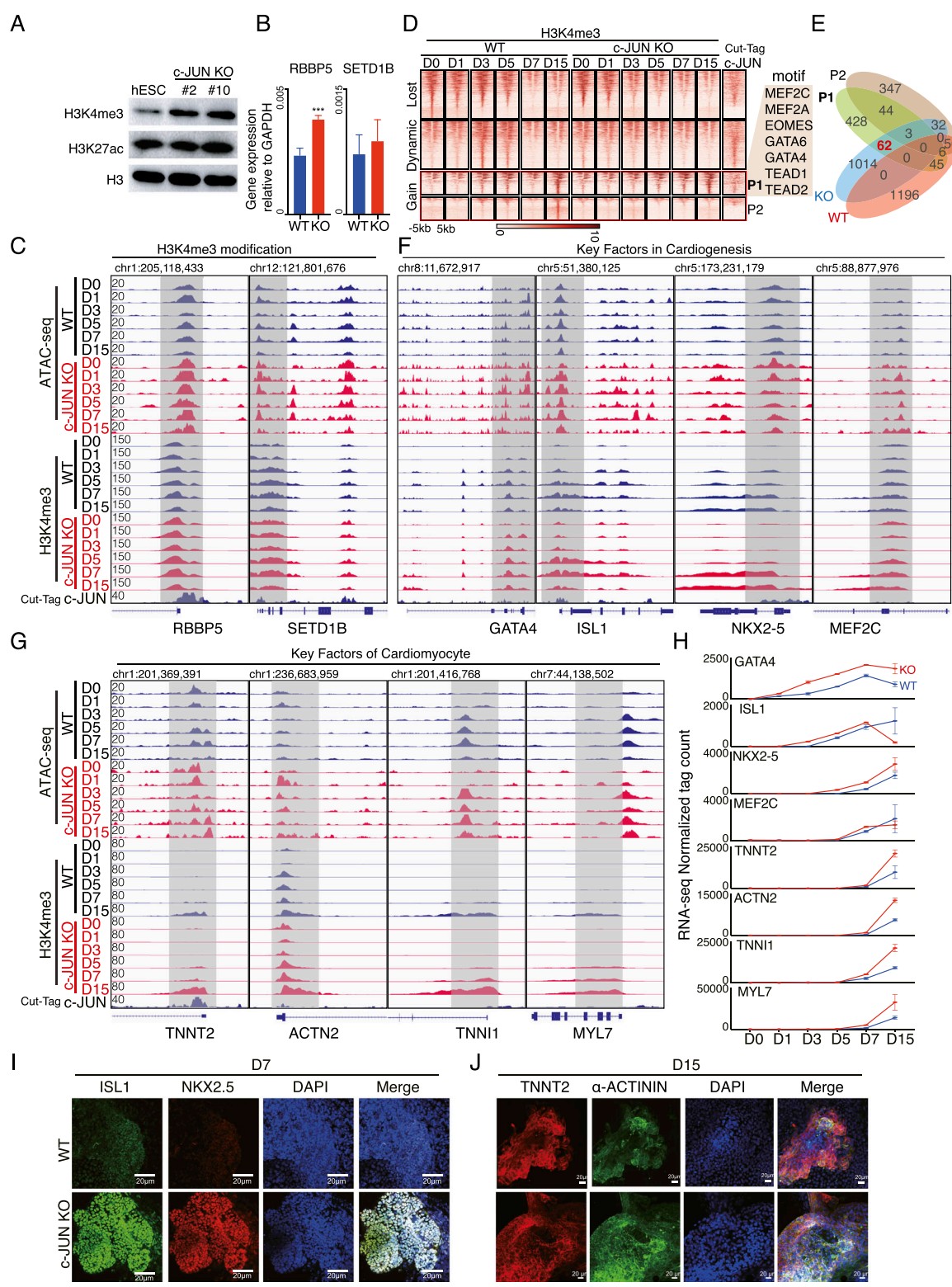

**Figure 4. *c-JUN* inhibits H3K4me3 modification during cardiomyocyte generation.**
**(A)** Western blot showing the levels of H3K4me3 and H3K27ac in WT and knockout *c-JUN* hESC. **(B)** Relative expression of the indicated SET complex genes in WT and *c-JUN* KO hESC. Data were from eight biological replicates in four independent experiments and are shown as the mean ± SEM. *** *P*-value < 0.001, Unpaired *t* test between the WT and KO (*c-JUN* KO) groups. **(C)** Genome view of the ATAC-seq data, H3K4me3 ChIP-seq, and c-JUN CUT&Tag data at the SET complex genes *RBBP5* and *SETD1B* loci. **(D)** Heatmap showing the read density for H3K4me3 ChIP-seq data during the time course of WT and *c-JUN* KO hESC differentiation into cardiomyocytes. c-JUN binding profile was generated from hESC by CUT&Tag. The data were arranged into Lost, Dynamic, and Gain categories, based on the behavior of H3K4me3 in WT cells. When

To explore how *c-JUN* affects H3K4me3 modification, we performed qRT-PCR to detect the expression of genes that are part of the H3K4 methyltransferase SET/MLL complex. Data showed that the knockout of *c-JUN* promoted the expression of *RBBP5* and *SETD1B*, core components of the SET complex (33) (Figs 4B and S2A and B). Cleavage under targets and tagmentation (CUT&Tag) data also showed c-JUN is bound to the promoter regions of *RBBP5* and *SETD1B* (Fig 4C). This suggests that *c-JUN* represses SET complex activation by inhibiting *RBBP5* and *SETD1B* expression, thus influencing the deposition of H3K4me3.

To gain a genome-wide view of the H3K4me3 modification dynamics, we performed ChIP-seq for H3K4me3 and clustered the peaks into three categories. Loci with H3K4me3 at the early stage but lost during cardiogenesis (Lost), loci only with H3K4me3 at the middle stage (Dynamic), and loci that gain H3K4me3 at the late stage (Gain) (Fig 4D). We further subdivided the Gain group into P1 (those that had gained H3K4me3 in both WT and KO cells) and P2 (loci that had no H3K4me3 in the *c-JUN* KO cells) (Fig 4D). Interestingly, the Gain H3K4me3 modification loci (P1) are also enriched with key TFs that relate to cardiogenesis such as *MEF* families, *GATA* families, and *TEAD* families (Fig 4D). To explore whether *c-JUN* represses cardiomyocyte differentiation by regulate H3K4me3 modification, we analyzed the DEGs associated with P1 and P2 loci. In total, 62 genes were activated in *c-JUN* KO and had increased H3K4me3 (Fig 4E), including key factors involved in cardiogenesis, such as *GATA4*, *ISL1*, *NKX2-5*, and *MEF2C*, and key factors of cardiomyocytes, such as *TNNT2*, *ACTN2*, *TNNI1*, and *MYL7* (Fig 4F and G). In addition, these genes were higher expressed in the *c-JUN* KO cells (Fig 4H–J). Consistent with the motif analysis in Fig 3, *c-JUN* KO promoted the opening of loci enriched with DNA-binding motifs for *GATA*/*ISL1*/*NKX2-5*/*MEF2C*/*MEF2A* (Fig 3E and F).

Together, our data showed that c-JUN binds to and represses *RBBP5* and *SETD1B* expression, thus controlling H3K4me3 modification and chromatin accessibility dynamics and inhibiting the activation of the key factors which regulate cardiogenesis.

### Inhibition of H3K4me3 demethylation promotes cardiomyocyte differentiation

To investigate whether H3K4me3 modification itself is critical for cardiomyocyte differentiation, we used the small molecule CPI-455, a specific inhibitor of histone demethylases 5 (*KDM5*) (KDMi). Western blot showed that the level of H3K4me3 was increased during KDMi treatment (Fig 5A). And consistent with our hypothesis, when KDMi was added to the cardiomyocyte differentiation medium, the spontaneous contractions of cardiomyocytes in the KDMi group were observed to be stronger than those of control cells (Video 3 and Video 4). Flow cytometry analysis also recorded an increase in the cardiomyocytes marker TNNT2 in KDMi-treated cells

compared with the control (67% versus 56%; Fig 5B). Interestingly, the window experiment shows 7 d KDMi treatment (D0–D7) could generate a higher differentiation efficiency (Fig 5C and D). Immunofluorescence results showed that KDMi could increase TNNT2 protein levels and enhance the number of sarcomeres in cardiomyocytes (Fig 5E and F).

To validate whether KDMi could accelerate the key factors in regulating cardiogenesis, we performed a quantitative polymerase chain reaction (qRT-PCR) to detect cardiac mesoderm marker gene expression. The results show *TBXT* (*T*), *DKK1*, *EOMES*, *MEST*, *HAND2*, and *KDR*, key markers of cardiac mesoderm, were quickly activated on D1 or D3 in response to KDMi treatment (Fig 5G and H). These results indicate H3K4me3 plays an important role in cardiac mesoderm and cardiomyocyte development. Overall, either knocking out *c-JUN* or using a *KDM* inhibitor could up-regulate H3K4me3 and speed up cardiac mesoderm and cardiomyocyte formation (Fig 5I).

### Cell trajectory during hESC to cardiomyocytes differentiation

To uncover how *c-JUN* alters the cell trajectory of hESC to cardiomyocytes, we performed single-cell RNA-seq experiments and obtained 52,695 cells from hESC (D0), D3, D7, and D15 of WT and *c-JUN* knockout cells. In agreement with the bulk RNA-seq data, single-cell RNA-seq data also showed *TNNT2* was highly expressed in D7 and D15 in *c-JUN* KO conditions (Figs 4H and S3A). Knockout of *c-JUN* led to an enhanced percentage of TNNT2+ cells on D7 and a small decrease on D15 (Fig S3B). Potentially, the discrepancy between the single-cell RNA-seq data and FACS data may be caused by slower RNA transcription and reduced protein degradation at the cardiomyocyte stage. Next, we integrated the WT and *c-JUN* KO data sets and visualized the cell clusters through a t-SNE projection (Fig 6A). After cell type annotation, we found cell heterogeneity in D3 to D15 in both WT and *c-JUN* KO conditions (Figs 6A–C and S3C and D). Interestingly, knockout of *c-JUN* inhibited non-cardiomyocyte-related expression of *TNNT2* (Fig S3E). From the single-cell distributions, *c-JUN* KO and WT hESC clustered closely together, which suggests *c-JUN* KO hESC were similar to WT hESC, and *c-JUN* KO had only a minimal impact overall on hESC (Fig 6A). In agreement with previous results, the pluripotent marker *OCT4* was down-regulated quicker in KO than WT cells (Fig 6D). Cardiac mesoderm marker genes *TBXT* (*T*) and *HAND1* were activated earlier in KO cells, and *TNNT2* was highly expressed in both WT and KO cardiomyocytes (Fig 6D). For trajectory analysis, *c-JUN* KO and WT cells both appeared to develop three branches, including hESC to cardiomyocytes, epicardial, and endothelial cells (Fig 6E–G). Those results suggest *c-JUN* plays a role in regulating pluripotency (20) and support a role for *c-JUN* in inhibiting cardiomyocyte formation by repressing key factors involved in regulating cardiogenesis, in agreement with the H3K4me3 and ATAC-seq data.

---

overlapped with *c-JUN* KO data, the Gain H3K4me3 loci were further subdivided into P1 and P2 subgroups. P1 loci have H3K4me3 modification in both WT and *c-JUN* KO cells. P2 loci have no H3K4me3 modification in *c-JUN* KO cells. **(E)** Venn diagram showing the overlap of genes regulated by P1 and P2 loci with DEGs (defined in Fig 3I) in WT and *c-JUN* KO cells. **(F)** Genome view of the ATAC-seq data, H3K4me3 ChIP-seq, and c-JUN binding data for the key factors in cardiogenesis. **(G)** Genome view of the ATAC-seq data, H3K4me3 ChIP-seq, and c-JUN binding data for the key factors of cardiomyocytes. **(H)** The RNA expression pattern for the respective genes is shown in line plot. RNA-seq expression units are normalized tag counts. **(I, J)** Immunostaining showing the protein levels of the cardiac progenitor marker genes ISL1 and NKX2-5 (panel (I)) and cardiomyocyte marker genes TNNT2 and *α*-ACTININ (panel (J)). Scale bar, 20 *μ*m.
Source data are available for this figure.

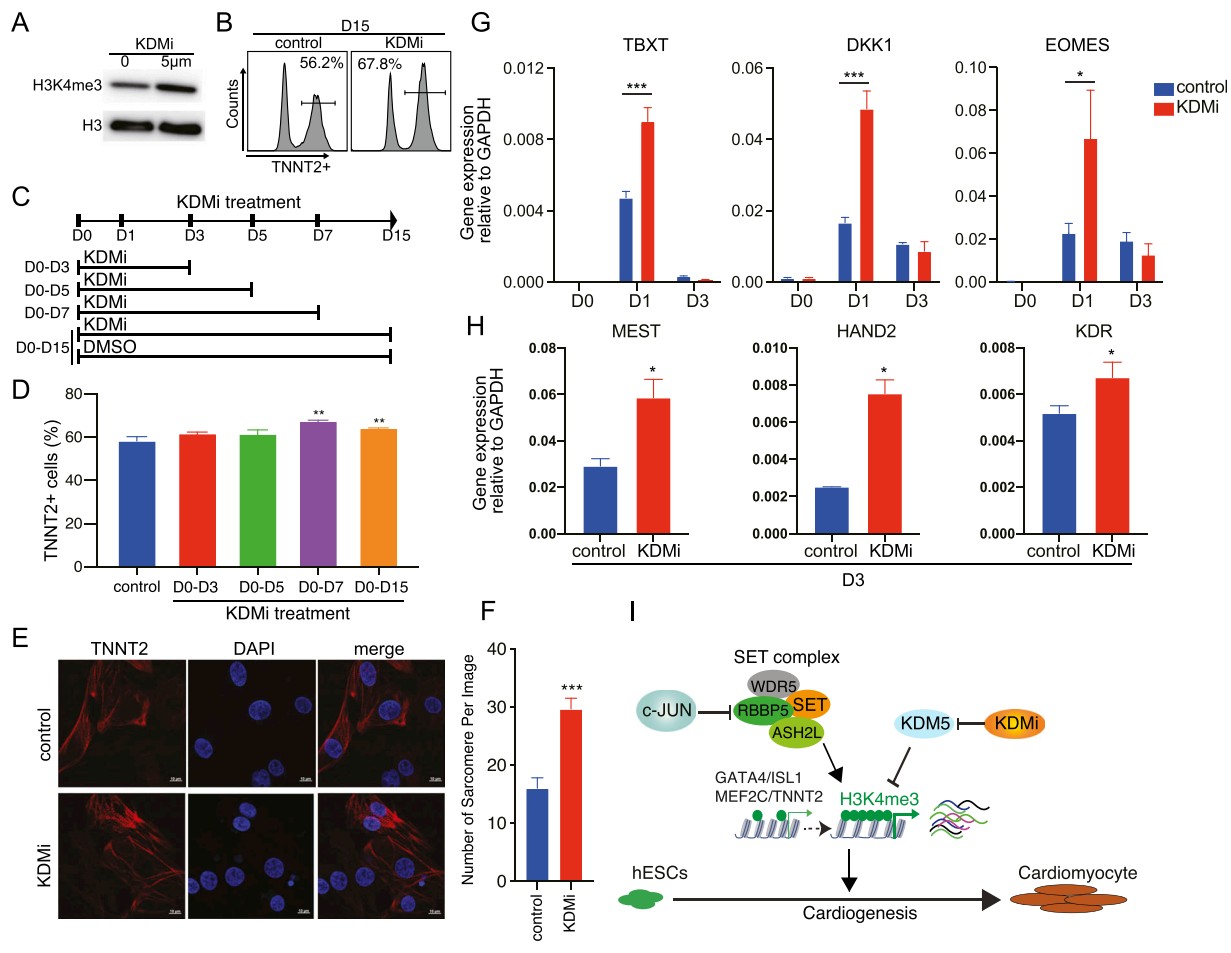

**Figure 5. KDM inhibitor promotes hESC differentiation into cardiomyocytes.**
**(A)** Western blot showing the H3K4me3 protein level in hESC treated with 0 μm and 5 μm KDMi. **(B)** Flow cytometry analysis of *TNNT2* expression in cardiomyocytes differentiated from hESC. Percent of cells TNNT2+ is indicated for each condition, control: induced cardiomyocytes from WT hESC with DMSO, KDMi: induce cardiomyocytes from hESC with DMSO-dissolved KDM inhibitor (CPI-455, 5 μm). KDMi was added into the culture medium from D0 to D15. **(C)** Schematic of the time window of KDMi treatment. **(C, D)** Flow cytometry quantification of cardiomyocyte differentiation efficiency, based on the percent of TNNT2+ cells in different groups in (C). Data were from eight biological replicates in three independent experiments and are shown as the mean ± SEM. ** *P*-value < 0.01, two-way ANOVA with Sidak correction between the control and KDMi groups. **(E)** Immunostaining showing the level of TNNT2 in cardiomyocytes that generated with DMSO or 7-D KDMi treatement (D0–D7). Scale bar, 10 μm. **(F)** Counts of the number of sarcomeres in each image. Data were from seven to nine replicates and are shown as the mean ± SEM. *** *P*-value < 0.001, two-way ANOVA with Sidak correction between the control and KDMi groups. **(G)** Relative expression of the indicated mesoderm marker genes during the differentiation of hESC into mesoderm cells with or without KDMi. Data were from six biological replicates in three independent experiments and are shown as the mean ± SEM. *** *P*-value < 0.001, * *P*-value < 0.05, two-way ANOVA with Sidak correction between the control and KDMi groups. **(H)** Relative expression of the indicated cardiac mesoderm marker genes during differentiation of hESC into cardiac mesoderm cells with or without KDMi. Data were from six biological replicates in three independent experiments and are shown as the mean ± SEM. * *P*-value < 0.05, Unpaired *t* test between the control and KDMi groups. **(I)** Schematic for the role of c-JUN in modulating chromatin accessibility and H3K4me3 modification during hESC to cardiomyocyte differentiation.
Source data are available for this figure.

### *c-JUN* inhibits key regulons activity during cardiogenesis

To investigate how *c-JUN* perturbs cardiomyocyte cell fate, we analyzed the gene regulatory network using SCENIC (34, 35). We analyzed the hESC-to-cardiomyocyte branch and used SCENIC to predict the TFs used in each cell type in WT and *c-JUN* KO conditions. The results showed 24 TFs, including *EOMES*, *HOX*, *GATA*, and *FOX* families, were enriched in the mesoderm-to-cardiomyocyte stages in *c-JUN* KO cells (Fig 6H). However, neural ectoderm genes *SOX9/SOX6* were enriched in WT cells. Consistent with the ATAC-seq data, WT cells opened loci related to

placental development (Figs 3E–G and 6H). Meantime, H3K4me3 modification loci were also enriched with *EOMES/GATA/MEF* TFs that are predicted by SCENIC, indicating transcriptome, epigenetic modification, and chromatin dynamics were synergistic remodeling to reshape the cell fate to cardiomyocyte (Fig 4D). These results illustrate the differential gene regulatory network used in WT and *c-JUN* KO cells and indicate *c-JUN* plays an important role in cardiac mesoderm and cardiomyocyte cell fate decisions.

Together, our data suggest that *c-JUN* controls cardiogenesis by inhibiting chromatin opening and repressing the expression

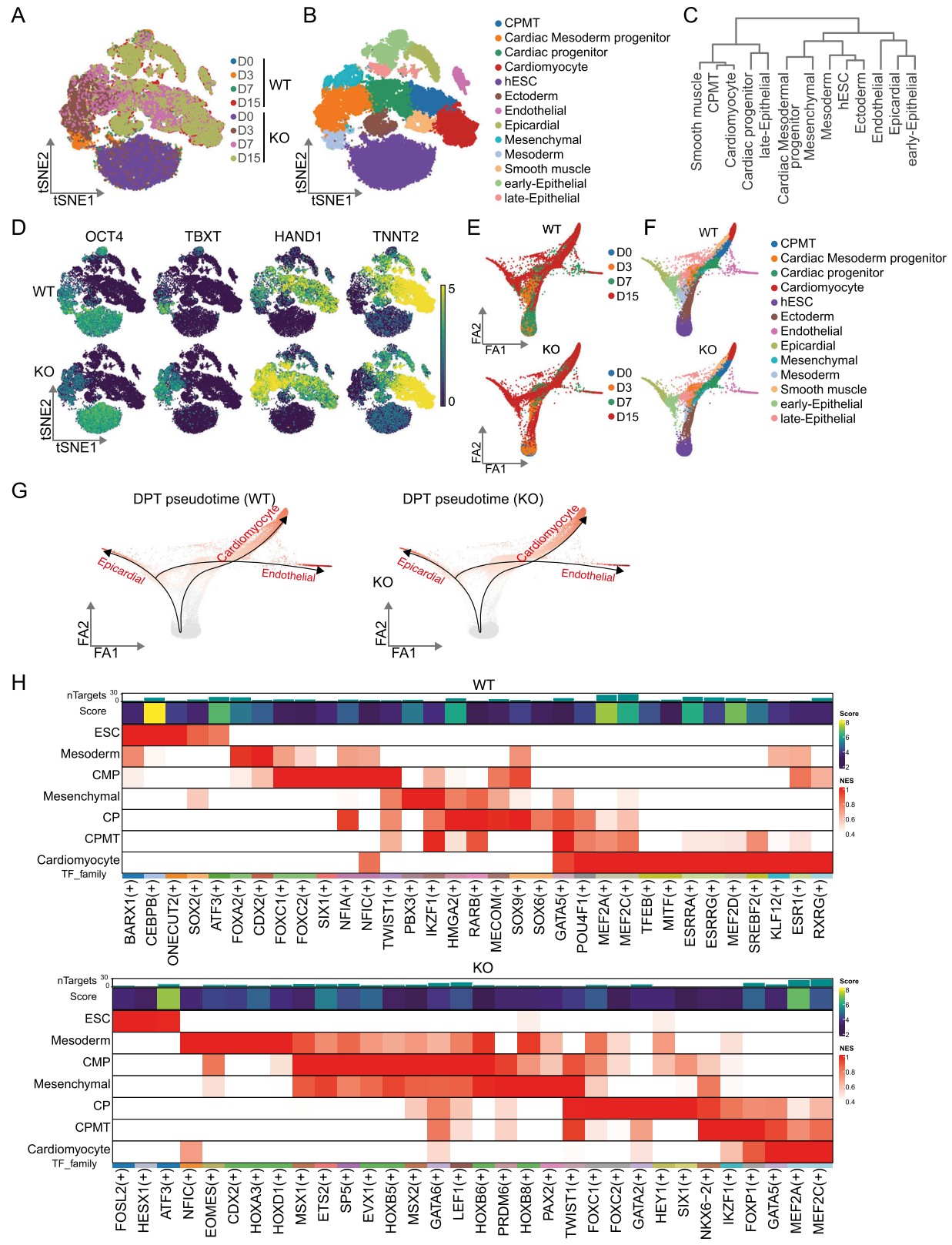

**Figure 6.   Single-cell transcriptome analysis showing the trajectories of cardiomyocyte differentiation.**
**(A, B)** A t-SNE showing the single-cell RNA-seq data of WT and *c-JUN* KO cells during hESC differentiation into cardiomyocytes. **(B)** 13 cell clusters were defined according to the specific marker genes. **(C)** Dendrogram plot shows the correlation of the 13 cell clusters. **(D)** t-SNE colored by the expression level of representative marker genes during the induction of hESC into cardiomyocytes. Each dot represents one cell, and RNA level is colored by selected marker intensities from high (yellow) to low (purple).

of key factors that regulate cardiogenesis and cardiomyocyte formation. Furthermore, *c-JUN* was bound to the loci of and repressed the expression of the SET complex genes *RBBP5* and *SETD1B*, thus leading to a lower level of H3K4me3 on the promoters of key cardiogenesis factors to inhibit cardiomyocyte differentiation.

## Discussion

*c-JUN* was previously reported as a key TF for embryonic development (15, 16). During mouse embryonic development, knockout of *c-Jun* leads to the failure to form a correct heart structure (36). However, the mechanism of *c-JUN* in embryonic development remains unknown, especially in humans. In this work, we used an in vitro hESC-to-cardiomyocyte differentiation system to study *c-JUN*'s function and found that *c-JUN* acts as a barrier to cardiomyocyte differentiation. The small molecular JNK inhibitor (SP600125) could phenocopy the *c-JUN* KO phenotype and release the negative impact of *c-JUN* in cardiomyocyte generation (Fig 2F). Further study revealed that *c-JUN* inhibits the hESC-to-cardiomyocyte transition by maintaining the chromatin state and affecting the modification of H3K4me3 by inhibiting SET complex activation.

In agreement with the increased H3K4me3 level by knocking out *c-JUN* to promote cardiomyocyte differentiation, manipulating H3K4me3 modification by KDMi also activated mesoderm/cardiac mesoderm genes, thus constantly pushing the cells toward a CM cell fate. Recently, a study published in Exp Mol Med suggested that inhibiting *KDM5A* could ameliorate pathological cardiac fibrosis (37). However, another study submitted to BioRxiv (38 *Preprint*) suggests *KDM5A* could regulate cardiomyocyte maturation by promoting fatty acid oxidation, oxidative phosphorylation, and myofibrillar organization. Those studies indicate a complex role for *KDM5* in regulating cardiomyocyte cell fate decision and maturation.

Our study contradicts previous in vivo studies of the *c-Jun* KO in mice which were post-implantation lethal at around E13.5 because of cardiac problems (36). Our work shows, conversely, that *c-JUN* is a barrier to cardiomyocyte differentiation. However, it is important to highlight critical differences between the in vivo knockout of *c-Jun* and the in vitro differentiation system used here. Potentially, an accelerated generation of cardiomyocytes may be deleterious for proper heart generation because of reduced numbers of support cells. Indeed, our single-cell RNA-seq data indicate that the cardiomyocyte population expands at the expense of epicardial cells. Potentially, in the even more complex in vivo development of the heart, other cell types may also be defective, leading to a failure to establish the correct balance of cells required for a functioning heart organ.

*c-JUN* is a somatic key factor and could repress somatic cell reprogramming and induce pluripotency exit quickly when overexpressed in mouse ESC (39). However, in human ESC, *c-JUN* helps maintain the hESC in a pluripotent state; these contradictory characteristics of *c-JUN* may be caused by the context-dependent nature of *c-JUN*, as in somatic cells, *c-JUN* co-binds with TFs to inhibit cell fate transitions (27, 40), but in hESC, *c-JUN* is co-binding with pluripotent TFs to inhibit differentiation (20). This conflicting role of *c-JUN* in different cell types shows how *c-JUN* can function as a guardian to maintain cell fate identity.

Our work demonstrates how *c-JUN* can impact chromatin accessibility, and inhibit cell fate transitions (27). Here, we use the cardiomyocyte generation model to study *c-JUN*'s function and find that *c-JUN* could inhibit chromatin opening and limit chromatin accessibility. This showed that *c-JUN* has a powerful ability to modulate chromatin structure. Previous studies have shown that *c-JUN*/*AP-1* could interact with cell type-specific TFs and bind to nucleosomes to facilitate recruitment of BAF complex and increase chromatin accessibility (40). We also found *c-JUN* could regulate H3K4me3 by binding to and repressing *RBBP5* and *SETD1B* transcriptional activation, the core components of the H3K4 methyltransferase SET complex. Indeed, H3K4me3 is an open (active) chromatin marker that has been reported to play a key role in mesoderm and cardiomyocyte cell fate decisions (29, 41) and even play an important role in the development of cardiovascular disease (42, 43, 44). Overall, our work uncovered a new function of *c-JUN* in regulating chromatin accessibility and H3K4me3 modification which play an important role in cardiomyocyte cell fate decision.

Finally, *c-JUN*/*AP-1* families play complex roles in embryonic development (15, 16), tissue regeneration (19), somatic cell reprogramming, hESC differentiation (20, 27, 39), and chromatin structure remodeling (27, 40, 45, 46). However, the role of *c-JUN* in cell fate decisions and epigenetic regulation remains unclear; indeed, opposing roles for *c-JUN* have been reported in some conditions (20, 39). For a more comprehensive study of the function of *c-JUN*, new model systems such as early embryonic development and new methods such as single-cell ATAC-seq and spatial single-cell technology should be used to explore the novel role of *c-JUN* in cardiomyocyte cell fate decisions.

## Materials and Methods

### Differentiation of human ESC to cardiac

Differentiation toward cardiomyocyte lineage was induced using small molecules to modulate the WNT signaling pathway and was optimized according to a small molecule-based monolayer method as previously described (21). Briefly, hESC cultured in mTeSR1 medium were digested into single cells by Accutase (Sigma-Aldrich) and plated onto Matrigel-coated culture dishes at a density of 4 x $10^5$ cells/12 well in mTeSR1 medium with ROCKi inhibitor (Y-27632; Selleck) for 24 h. After 1 d, Y-27632 was withdrawn from the medium,

---

**(E, F)** Cell differentiation trajectory calculated using the ForceAtlas2 algorithm. **(E, F)** WT and *c-JUN* KO cells are shown in separated plots (E) based on time and (F) based on the cell types. **(G)** Diffusion pseudotime shows the differentiation trajectory of WT and *c-JUN* KO cells. **(H)** SCENIC results on the WT and *c-JUN* KO condition, master regulators are color-matched with the cell types.

and Cells were then induced with mTeSR1 medium for 48 h. On day 0, cells were treated with cardiac differentiation medium for 24 h, consisting of RPMI/1640 and CHIR99021 (4 $\mu$M; Selleck). On day 1, the medium was exchanged by fresh RPMI/1640, and cells were induced for another 48 h. Subsequently, on day 3, the medium was changed to RPMI/1640 medium containing IWP2 (5 $\mu$M; Selleck) for 2 d. After 2 d (day 5), the medium was exchanged for RPMI/1640 medium for another 2 d. For maintenance of the obtained cardiomyocytes, at day 7, the medium was changed to RPMI/1640 containing insulin (10 $\mu$g/ml) and Vitamin C (200 $\mu$g/ml), and half the medium was exchanged every day until day 15. Contracting cells were seen from day 9 to day 15.

### Generation of knockout hESC

*c-JUN-/-*hESC was generated by CRISPR/Cas9, and pairs of sgRNA were used to delete the target exons. sgRNA1: acaagtttcggggccg-caac; sgRNA2: gagaacttgacaagttgcga.

### Flow cytometry

Cells were dissociated with Accutase (Sigma-Aldrich) and filtered through a 70-$\mu$m cell strainer (BD Biosciences) to get a single-cell suspension. Single cells were fixed with 4% paraformaldehyde for 20 min and washed twice in PBS for at least 5 min each. Then, cells were incubated in 1% BSA for 30 min, permeabilized in 0.2% Triton X-100 for 10 min, and washed three times with PBS containing 0.05% Triton X-100. Subsequently, cells were stained with 647-conjugated primary antibodies (1:200; cTnT) for 1 h at room temperature. After PBS washing steps, cells were analyzed using an Accuri C6 Plus flow cytometer (BD Biosciences), and data were processed using FlowJo 10.4 software.

### Immunofluorescence

Cells growing on coverslips were washed three times with PBS, then fixed with 4% PFA for 30 min, and subsequently, cell membranes were penetrated with 0.1% Triton X-100 and blocked with 3% BSA for 30 min at room temperature. Cells were then incubated with c-JUN antibody for 1 h. After three washes in PBS, followed by 1 h of incubation in secondary antibodies, cells were incubated in DAPI for 1 min. Then the coverslips were mounted on the slides for observation on a confocal microscope (Leica). The following antibody was used in this project: anti-c-JUN (no. 9165, 1:100; CST).

### Western blot analysis

hESC or differentiated cardiomyocytes were lysed in RIPA buffer (Thermo Fisher Scientific). The extracted proteins were quantified by a BCA Protein Assay Kit (Thermo Fisher Scientific), respectively. Total protein was subjected to SDS–PAGE, then transferred to a polyvinylidene fluoride membrane (PVDF) and probed with the specific primary antibodies at 4°C overnight. The PVDF membrane was washed with PBST three times, and then the HRP-conjugated secondary antibody was incubated for 1 h at room temperature.

Proteins were detected by enhanced chemiluminescent substrates (ECL Kit; Thermo Fisher Scientific). The antibodies used are shown in Table S1.

### ATAC-seq

ATAC-seq was performed as previously described (25). Briefly, 50,000 cells were washed with 50 $\mu$l cold PBS and resuspended in 50 $\mu$l lysis buffer (10 mM Tris–HCl pH 7.4, 10 mM NaCl, 3 mM MgCl2, 0.2% [vol/vol] IGEPAL CA-630). The suspension of nuclei was then centrifuged for 10 min at 500$g$ at 4°C, followed by the addition of 50 $\mu$l transposition reaction mix (25 $\mu$l TD buffer, 2.5 $\mu$l Tn5 transposase, and 22.5 $\mu$l nuclease-free H$_2$O) from the Nextera DNA library Preparation Kit (96 samples) (FC-121-1031; Illumina). Samples were then PCR amplified and incubated at 37°C for 30 min. DNA was isolated using a MinElute Kit (QIAGEN). ATAC-seq libraries were first subjected to five cycles of pre-amplification. To determine the suitable number of cycles required for the next round of PCR, the libraries were assessed by quantitative PCR as described (26), and the libraries were then PCR amplified for the appropriate number of cycles according to the qRT-PCR results. Libraries were purified with a Qiaquick PCR (QIAGEN) column, and the library's concentration was measured using a KAPA Library Quantification Kit (KK4824) according to the manufacturer's instructions. Finally, the ATAC libraries were sequenced on the NextSeq 500 sequencing platform using a NextSeq 500 High Output Kit v2 (150 cycles) (FC-404-2002; Illumina) according to the manufacturer's instructions.

### ChIP-seq

H3K4me3 ChIP was performed as described previously (39). Briefly, 1 × 109 cells were fixed in 8.75 ml DMEM/F12 with 1% formaldehyde for 15 min at room temperature with rotation and then followed by the reaction with 0.125 M glycine. Cells were then lysed in ChIP-buffer A (50 mM HEPES-KOH, 140 mM NaCl, 1 mM EDTA [pH 8.0], 10% glycerol, 0.5% NP-40, 0.25% Triton X-100, 50 mM Tris–HCl [pH 8.0], and protease inhibitor cocktail) for 10 min at 4°C. Samples were centrifuged at 1,400$g$ for 5 min at 4°C. Pellets were lysed in ChIP-buffer B (1% SDS, 50 mM Tris–HCl [pH 8.0], 10 mM EDTA, and protease inhibitor cocktail) for 5 min at 4°C. The DNA was fragmented to 100–500 bp by sonication and then centrifuged at 12,000$g$ for 2 min. The supernatant was diluted with ChIP IP buffer (0.01% SDS, 1% Triton X-100, 2 mM EDTA, 50 mM Tris–HCl [pH 8.0], 150 mM NaCl, and protease inhibitor cocktail). Immunoprecipitation was performed with 2 $\mu$g rabbit anti-H3K4me3 antibody (ab8580; Abcam) coupled to Dynabeads with protein A/G overnight at 4°C. Beads were washed, eluted, and reverse cross-linked. DNA was extracted with phenol/chloroform for sequencing. The ChIP DNA library was constructed with VAHTS Universal DNA Library Prep Kit for Illumina (Nd604; Vazyme Biotech) according to the manufacturer's instructions. The DNA libraries were quantified and tested by qRT-PCR with positive primers to assess the quality of the library. Then, libraries were sequenced on an Illumina NextSeq 500 instrument using 75-bp paired-end reads.

## CUT&Tag

c-JUN CUT&Tag data was generated by the Hyperactive Universal CUT&Tag Assay Kit (Vazyme) according to the manufacturer's instructions. H1 cells ($1 \times 10^5$) were used for the CUT&Tag experiment, and pA-Tn5 transposase was used to cut the genome and add a special adaptor sequence to build a library. The enrichment of target sites in the library was detected using qPCR. The c-JUN antibody was purchased from Cell Signaling Technology (#9165; Cell Signaling Technology).

## Single-cell sequencing

Cells at hESC (D0) and differentiated to cardiomyocyte (D3, D7, and D15) time points were collected and resuspended in DPBS with 0.04% BSA. Then, cell suspensions (500–1,000 cells/µl) were loaded on a Chromium Single Cell Controller (10x Genomics) to obtain single-cell gel beads in emulsion (GEMs) by using Single Cell 3' Library and Gel Bead Kit V2 (120237; 10x Genomics). Captured cDNAs were lysed, and the released RNA was barcoded through reverse transcription in singular GEMs. Barcoded cDNAs were pooled and cleaned by DynaBeads MyOne Silane Beads (37002D; Invitrogen). Single-cell RNA-seq libraries were prepared by Single Cell 3' Library Gel Bead Kit V2 (120237; 10x Genomics) following the manufacturer's instructions. Sequencing was operated on an Illumina HiSeq X 10 with pair-end of 150 bp (PE150).

## RNA-seq and gene expression analysis

Total RNA was prepared with TRIzol. For quantitative PCR, cDNAs were synthesized with ReverTra Ace (Toyobo) and oligo-dT (Takara) and then analyzed by qRT-PCR with Premix Ex Taq (Takara). For RNA-seq, TruSeq RNA Sample Prep Kit (RS-122-2001; Illumina) was used for library construction, and the sequencing was performed using a NextSeq 500 High Output Kit v2 (75 cycles) (FC-404-1005; Illumina), according to the manufacturer's instructions. RNA-seq was processed as previously described (47, 48); briefly, reads were aligned to a transcriptome index generated from the Ensembl annotations, using RSEM (49), Bowtie2 (50), and normalized using EDASeq (51). RNA-seq data are expressed in units of GC-normalized tag counts. DEGs were analyzed by DESeq2 (52).

## ATAC-seq bioinformatic analysis and peak calling

All sequencing data were mapped to the hg38 human genome assembly using Bowtie2 with the options –very-sensitive. samtools was used to remove low-quality mapped reads with option view –q 35, and unique reads were kept. We removed mitochondrial sequences using "grep –v 'chrM.'' Biological replicates were merged, and peaks were called using dfilter (53) (with the settings: –bs = 100 –ks = 60 –refine –std = 5). BigWig files were produced using genomeCoverageBed from bedtools (scale=107/<one sample's total unique reads>) and then bedGraphToBigWig. Gene ontology and gene expression measures were called by first collecting all transcription start sites within 10 kb of an ATAC-seq peak and then performing GO analysis with goseq (54), or measuring gene expression. Other analysis was performed using GLBase (47).

## Recalling weak peaks from the ATAC-seq data

Peak recalling is based on the method we previously described (27). Briefly, when dfilter is used to discover peaks, as described above, it is generally conservative and will not call a weak peak. Hence, we "re-call" all peaks by measuring the sequence tag density in all ATAC-seq libraries for all possible peaks in any other ATAC-seq library, irrespective of which library the peak was called in by dfilter (53). Then we get a superset of all possible peaks in any library, and based on our previous analysis, we used an arbitrary minimum threshold of 0.2734 to filter out false peaks; if the ATAC-seq is below this value, it is annotated as "closed" and above "open." All downstream analysis was based on this new peak list.

## ChIP-seq data analysis

Reads from ChIP-seq experiments were mapped to the mouse genome (hg38) using Bowtie2 (–very-sensitive), as described for ATAC-seq data, and only the uniquely mapped reads were kept for further analysis. Peaks were called using MACS2 (55) software with the default parameters.

## Computational analysis of TF binding sites and open chromatin regions

Motif analysis was performed by HOMER 28 with default settings. Motifs were only kept if the *P*-value was <0.01 and (<percent of target>/<percent of background>) was >1.5. Annotation of the ChIP-seq/ATAC-seq peaks/open chromatin to genes was performed by HOMER using default settings.

## Single-cell RNA-seq data analysis

Fastq reads were aligned to the genome using STAR (56) with the setting ' –soloType Droplet –soloFeatures Gene –runThreadN 20 –soloCBstart 1 –soloCBlen 16 –soloUMIstart 17 –soloUMIlen 12 –soloBarcodeReadLength 0 –readFilesCommand zcat –outSAMtype BAM SortedByCoordinate –outSAMattributes NH HI AS nM CR CY UR UY'. The count matrix was lightly filtered to exclude cell barcodes with low numbers of counts: Cells with <2,000 UMIs, <500 genes, or >20% fraction of mitochondrial counts were removed. The filtered matrix was normalized according to Scanpy tutorials. The top 3,000 most highly variable genes were used for principal component analysis, and the first six PCs (principal components) were used for downstream analysis with Scanpy. Gene regulatory network inference with SCENIC was used (35).

# Data Availability

The raw sequence data reported in this work have been deposited in the Genome Sequence Archive (Genomics, Proteomics & Bioinformatics 2021) in the National Genomics Data Center (Nucleic Acids Res 2021), China National Center for Bioinformation/Beijing Institute of Genomics, Chinese Academy of Sciences (GSA: HRA003227) that are publicly accessible at https://ngdc.cncb.ac.cn/gsa.

# Supplementary Information

# Acknowledgements

This work was supported in part by the National Natural Science Foundation of China (92068201, 32270574, 32150710521), the National Key Research and Development Program of China (2018YFA0108700 and 2017YFA0105602), the Guangdong Science and Technology Project (2020B1212060052, 2022B1212010010), and Science and Technology Projects in Guangzhou (2023A03J0045, 2023A04J0156). Guangzhou Key Laboratory of Biological Targeting Diagnosis and Therapy (202201020379) and the Youth Innovation Promotion of the Chinese Academy of Sciences (2019348) to D Li. Guangdong Cardiovascular Institute (2020XXG002), High-level Hospital Construction Project (DFJHBF202110, DFJHBF202111), and Plan on Enhancing Scientific Research in GMU.

## Author Contributions

H Zhong: validation and methodology.
R Zhang: resources, validation, methodology, and project administration.
G Li: resources and validation.
P Huang: resources and validation.
Y Zhang: resources and validation.
J Zhu: resources and validation.
J Kuang: resources.
AP Hutchins: resources.
D Qin: resources and funding acquisition.
P Zhu: funding acquisition.
D Pei: funding acquisition.
D Li: conceptualization, data curation, formal analysis, supervision, funding acquisition, validation, investigation, visualization, methodology, project administration, and writing—original draft, review, and editing.

## Conflict of Interest Statement

The authors declare that they have no conflict of interest.

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
