## [Reviewer comments · Life Science Alliance]

Life Science Alliance

c-JUN is a barrier in hESC to cardiomyocyte transition

Dongwei Li, Hui Zhong, Ran Zhang, Guihuan Li, Ping Huang, Yudan Zhang, Jieying Zhu, Junqi Kuang, Andrew Hutchins, Da Qin, Ping Zhu, and Duanqing Pei

DOI: <https://doi.org/10.26508/lsa.202302121>

Corresponding author(s): Dongwei Li, Guangzhou Medical University; Duanqing Pei, Westlake University; and Ping Zhu, Guangdong Provincial People's Hospital, Guangdong Academy of Medical Sciences

Review Timeline:

Submission Date:	2023-04-27
Editorial Decision:	2023-05-30
Revision Received:	2023-07-12
Editorial Decision:	2023-08-04
Revision Received:	2023-08-06
Accepted:	2023-08-07

Scientific Editor: Novella Guidi

Transaction Report:

May 30, 2023

Re: Life Science Alliance manuscript #LSA-2023-02121-T

Dr. Dongwei Li
Guangzhou Medical University
China

Dear Dr. Li,

Thank you for submitting your manuscript entitled "c-JUN is a barrier in hESCs to cardiomyocyte transition" to Life Science Alliance. The manuscript was assessed by expert reviewers, whose comments are appended to this letter. We invite you to submit a revised manuscript addressing the Reviewer comments.

Thank you for this interesting contribution to Life Science Alliance. We are looking forward to receiving your revised manuscript.

Sincerely,

B. MANUSCRIPT ORGANIZATION AND FORMATTING:

Reviewer #1 (Comments to the Authors (Required)):

Zhong et al. have described an in vitro cardiomyocyte differentiation system using human embryonic stem cells (hESCs) and evaluated the functional role of c-JUN, a TF belonging to AP-1 protein family, during cardiogenesis. The authors have revealed that, instead of playing a key role during early mouse embryo development, c-JUN functions as a barrier of cardiomyocyte formation. They further investigated how c-JUN inhibits the cardiomyocyte differentiation and showed that loss of c-JUN increases RBBP5 and SETD1B, two methyltransferases, and thus increases the enrichment of H3K4me3 at cardiac gene loci. The manuscript is of interest, but descriptive to some extent. A few key issues need to be fixed before the manuscript could be further considered for publication.

- The author mentioned many times in the manuscript that c-JUN "binds" to RBBP5/SETD1B and thus represses their expression. I agree RBBP5 and SETD1B are up-regulated upon c-JUN depletion. Then, experimental result such as ChIP, cut&tag (though there is cut & tag method in the M&M section and a heatmap only in Fig. 4D), or EMSA to show c-JUN really binds to or localizes at RBBP5/SETD1B loci.

- As the author claimed c-JUN is a barrier of cardiogenesis, I am wondering what would happen when overexpressing c-JUN during cardiomyocyte differentiation.

- Did the authors quantify the numbers of TNNT2 positive cells in WT and c-JUN scRNA-seq? I am wondering whether the authors also see more TNNT2 positive "cardiac cells" in KO compared to WT from scRNA-seq data.

- I think the authors should further discuss the contradictory phenomenon of c-JUN deletion observed between human and mouse early development.

- The whole manuscript suffers from grammar issues, mistakes, and typos. It is highly recommended to have the text proofread by someone more skilled with scientific writing.

Reviewer #2 (Comments to the Authors (Required)):

The manuscript from Zhong, Zhang, and Li et al. present work describing a role for c-Jun in cardiomyocyte differentiation repression. The c-Jun knockout cells (or c-Jun inhibitor treated) produce more cardiomyocytes than the controls. The authors go on to show that c-Jun mutant cell lines have higher H3K4me3 signals than control lines at cardiomyocyte related genes. Using single cell RNA-seq, the authors describe that c-Jun mutant lines have more cardiomyocyte-related gene expression. The work presented here is describes a potential role for c-Jun in inhibiting some cells from undergoing cardiomyocyte differentiation but seems preliminary in its proposed mechanism.

Below are some comments and questions related to the work presented in no particular order:

1. In the section, "Chromatin dynamics during cardiogenesis," (lines 161-177), it is unclear from the text that the authors are describing chromatin accessibility near the example genes and not something else like motif analysis.
2. The ATAC-seq and RNA-seq analysis in Figure 3 is performed on bulk cells undergoing differentiation. Based on the results from Figure 2D and Figure 6D, it seems reasonable that the WT cell differentiation contains a mixture of cell fates and states. On the other hand, Figure 2D suggests that the c-Jun mutant cell lines represent a more homogenous cardiomyocyte population (or at least, 92-96.5% of cells are TNNT2+). The authors statement about c-Jun mutants having more/less accessibility at certain loci or increased/decreased gene expression of certain regulators seems more to reflect the nature of the pool of cells (more cardiomyocytes, less other cell fates or states) rather than the behavior or action of c-Jun in any given cell within the pool of cells.
3. A similar issue as my comment above (#2), the H3K4me3 ChIP-seq analysis is performed on bulk cells undergoing differentiation. While it is certainly intriguing that global levels of H3K4me3 were up in the cJun KO lines compared to controls, it seems premature to conclude that H3K4me3 levels were increased at specific loci since the WT differentiation is a mixture of cell types at each time point compared to a seemingly more homogenous pool in the cJun mutant differentiations.
4. In Figures 2G and 5C, the authors state that the cardiomyocytes have enhanced sarcomere structures. How was this

- determined or quantified? Likewise, in Movies EV3 and S4, how was beating strength measured or quantified?
5. From the single cell analysis, what were the distribution of the 13 clusters between WT and cJun mutant cells? It was reported in Figure 2 that almost all cells in the cJun mutant population at Day 15 were TNNT2+ cardiomyocytes, but it seems like in 6E many Day 15 mutant cells are epicardial, perhaps even more epicardial than WT differentiation (Figure 6D).
 6. Is there any explanation for why the KDM5 inhibitor would have the specific effect of activating cardiomyocyte genes and not other cell fates?
 7. Lines 224 and 306, the authors state that they utilized ChIP-seq to demonstrate c-Jun binding, but the methods suggest they used a different method, namely, CUT&Tag.
 8. Why were clones #2 and #10 chosen over any of the other clones? No details are provided in the text.
 9. The manuscript and figures contained typos and grammatical errors that I would recommend fixing. Below are a few that jumped out to me during the review process:
 - a. Figure 1A, "Cardiac Mesoderm"
 - b. Figure 4C and 4G, it should say ATAC-seq along the lefthand side
 - c. Line 125, it should read, "we analyzed the time course RNA-seq data"
 - d. Line 138-139, it should read, "c-Jun was expressed throughout hESC"
 - e. Line 140, it should read, "and used CRISPR/Cas9 to delete"
 - f. Line 168, "MESP1"
 - g. Line 180, "chromatin accessibility is driven"
 - h. Line 189, "we compared the"

Reviewer #3 (Comments to the Authors (Required)):

c-JUN has been previously identified as a critical TF for mouse embryonic development. In this manuscript, the authors discovered that c-JUN plays an important role in cardiomyocyte development using in vitro differentiation of human embryonic stem cells (hESCs). Loss of c-JUN leads to increased expression of RBBP5 and SED1B, thereby improving chromatin accessibility and deposition of H3K4me3 on regulatory elements associated with cardiomyocyte development. This manuscript is somewhat innovative, as c-JUN is known to be essential for normal cardiac development in mice but acts as a barrier in hESCs to cardiomyocyte transition.

Major points:

1. Loss of c-JUN leads to early mouse embryonic death possibly due to failure to a normal cardiac system. develop. However, in this manuscript c-JUN is a barrier in hESCs to cardiomyocyte transition. There should be more explanation about the conflict function of c-JUN.
2. Are there any connections between the TFs predicted by SCENIN and H3K4me3? It would be valuable to supplement the discussion regarding the results shown in figure 6H.

Minor points :

1. The titles of video in the attachment do not align with the descriptions provided in the manuscript.
2. There are issues with the titles of Result1, Result2 and Result6, as they are not complete sentences. Furthermore, it is necessary to further standardize the language used in other sections of the manuscript
3. At line 158, the sentence "rather than a facilitator" lacks supporting results or references in the text.
4. How can we determine that "the differentiation trajectory of hESCs to cardiomyocytes transition was similar between WT and c-JUN KO conditions" from figure 3H.
5. The "h" in figure4 should be capitalized as "H". Additionally, are there RNA-seq data available for RBBP5 and SED1B as supplementary material?
6. There is some logical issue in lines 217-220. While chromatin accessibility can be regulated by histone methylation, the WB detection includes histone acetylation. This discrepancy should be addressed.
7. The description in line 224 is not very clear.
8. The font inconsistency in figure 5G should be addressed.
9. The color scheme used for WT and KO in figure 6A cannot be clearly distinguished.
10. The last sentence in line 278 has a weak correlation with the results presented in figure

Thank you for take care of my manuscript, here we have finished answering the question reviewers raised and we also finished the needed experiment that reviewer 1 asked. And as your mentioned before, I have putted the response to the reviewer's comments point by point after this page.

Thanks,
Sincerely,
Dongwei Li

Reviewer #1: Zhong et al. have described an in vitro cardiomyocyte differentiation system using human embryonic stem cells (hESCs) and evaluated the functional role of c-JUN, a TF belonging to AP-1 protein family, during cardiogenesis. The authors have revealed that, instead of playing a key role during early mouse embryo development, c-JUN functions as a barrier of cardiomyocyte formation. They further investigated how c-JUN inhibits the cardiomyocyte differentiation and showed that loss of c-JUN increases RBBP5 and SETD1B, two methyltransferases, and thus increases the enrichment of H3K4me3 at cardiac gene loci. The manuscript is of interest, but descriptive to some extent. A few key issues need to be fixed before the manuscript could be further considered for publication.

Response: We thank the reviewer for their comprehensive review of our paper, and the useful suggestions that we have utilized to improve our work.

1. The author mentioned many times in the manuscript that c-JUN "binds" to RBBP5/SETD1B and thus represses their expression. I agree RBBP5 and SETD1B are up-regulated upon c-JUN depletion. Then, experimental result such as ChIP, cut&tag (though there is cut & tag method in the M&M section and a heatmap only in Fig. 4D), or EMSA to show c-JUN really binds to or localizes at RBBP5/SETD1B loci.

Response: We apologize for not making this clearer. c-JUN is recruited to the *RBBP5* and *SETD1B* transcription start sites as shown in the CUT&Tag data in **Figure 4C** (the last row). We emphasize this data in the revision, and have revised the text to make the evidence for the claim explicit.

2. As the author claimed c-JUN is a barrier of cardiogenesis, I am wondering what would happen when overexpressing c-JUN during cardiomyocyte differentiation.

Response: This is an intriguing question. Our initial expectation is that overexpression of c-JUN would ultimately block cardiogenesis. To explore this we transfected a TetOn c-JUN plasmid into hESC and performed the cardiomyocyte differentiation experiment with and without 1 µg/ml dox to induce c-JUN overexpression. As the results shown over expression c-JUN extremely inhibit normal cardiomyocyte morphology and TNNT2+ cells (Box1). These results also added into Figure S1. (Lines 156-158)

experiments and are shown as the mean±SEM. *** p-value < 0.001, unpaired t test between the control and OE c-JUN groups.

3. Did the authors quantify the numbers of TNNT2 positive cells in WT and c-JUN scRNA-seq? I am wondering whether the authors also see more TNNT2 positive "cardiac cells" in KO compared to WT from scRNA-seq data.

Response: We quantified the numbers of TNNT2+ cells in WT and c-JUN KO scRNA-seq data (Box 2). When c-JUN was knocked out, TNNT2 was highly activated in D7 and D15 (A), also the proportion of TNNT2+ cells increased at D7 (~80%), followed by a small decrease at D15 (~70%). But in WT cells, TNNT2+ cells were ~60% at D7, and increased to ~80% on D15 (B). Potentially, the discrepancy between the single cell RNA-seq data and FACS data may be caused by slower RNA transcription and reduced protein degradation at the cardiomyocyte stage. Those results suggest that the knock out of c-JUN promotes TNNT2+ cell generation. In the revised version, we have added this result in Figure S3A and B to make the conclusion more rigorous.

4. I think the authors should further discuss the contradictory phenomenon of c-JUN deletion observed between human and mouse early development.

Response: Thanks for this comment. Our study contradicts previous *in vivo* studies of the c-Jun KO in mice which was post-implantation lethal at around E13.5 due to cardiac problems ¹. Our work shows, conversely, that c-JUN is a barrier to cardiomyocyte differentiation. However, it is important to highlight critical differences between the *in vivo* knock-out of c-Jun and the *in vitro* differentiation system employed here. Potentially, an accelerated generation of cardiomyocytes may be deleterious for proper heart generation due to reduced numbers of support cells. Indeed, our single-cell RNA-seq data indicates that the cardiomyocyte population expands at the expense of epicardial cells. Potentially, in the even more complex *in vivo* development of the heart, other cell types may also be defective, leading to a failure to establish the correct balance of cells required for a functioning heart organ. We have added the discussion about the contradictory phenomenon of c-JUN deletion between human and mouse early development in DISCUSSION section. (marked in blue color; Lines 349-359).

5. The whole manuscript suffers from grammar issues, mistakes, and typos. It is highly recommended to have the text proofread by someone more skilled with scientific writing.

Response: We have carefully checked the manuscript and also invited a native English writer to revise the manuscript.

Reviewer #2: The manuscript from Zhong, Zhang, and Li et al. present work describing a role for c-Jun in cardiomyocyte differentiation repression. The c-Jun knockout cells (or c-Jun inhibitor treated) produce more cardiomyocytes than the controls. The authors go on to show that c-Jun mutant cell lines have higher H3K4me3 signals than control lines at cardiomyocyte related genes. Using single cell RNA-seq, the authors describe that c-Jun mutant lines have more cardiomyocyte-related gene expression. The work presented here describes a potential role for c-Jun in inhibiting some cells from undergoing cardiomyocyte differentiation but seems preliminary in its proposed mechanism.

Response: We thank the reviewer for their comprehensive review of our paper and the useful suggestions to improve our work.

Below are some comments and questions related to the work presented in no particular order:

1. In the section, "Chromatin dynamics during cardiogenesis," (lines 161-177), it is unclear from the text that the authors are describing chromatin accessibility near the example genes and not something else like motif analysis.

Response: We agree this was not made clear. This part of the text was focused on chromatin dynamics at specific genes. To address this point we have revised the section to make it clearer, and have moved the motif analysis results (Fig. 3D) in 'Chromatin dynamics during cardiogenesis' section. (marked in blue color, Lines 181-190). This makes the presentation of the chromatin and motif analysis more unified.

2. The ATAC-seq and RNA-seq analysis in Figure 3 is performed on bulk cells undergoing differentiation. Based on the results from Figure 2D and Figure 6D, it seems reasonable that the WT cell differentiation contains a mixture of cell fates and states. On the other hand, Figure 2D suggests that the c-Jun mutant cell lines represent a more homogenous cardiomyocyte population (or at least, 92-96.5% of cells are TNNT2+). The authors statement about c-Jun mutants having more/less accessibility at certain loci or increased/decreased gene expression of certain regulators seems more to reflect the nature of the pool of cells (more cardiomyocytes, less other cell fates or states) rather than the behavior or action of c-Jun in any given cell within the pool of cells.

Response: We agree that this analysis can be difficult to interpret, However, as chromatin opening often precedes gene expression to reshape cell fate^{2,3}. We think it is reasonable to identify the sequence of chromatin changes that occurs in the WT and KO cells. Here, we found that KO of c-JUN resulted in increased accessibility at the chromatin of key TFs. This consequently regulates cardiomyocyte generation as early as in D3 (Box 3, Fig. 3B), which precedes the expression of TNNT2 (Box 2). This indicates that the loss of c-JUN primes chromatin in advance of changes in cell composition. and also explains why knock-out c-JUN could increase the population of TNNT2+ cells to 92%.

3. A similar issue as my comment above (#2), the H3K4me3 ChIP-seq analysis is performed on bulk cells undergoing differentiation. While it is certainly intriguing that global levels of H3K4me3 were up in the cJun KO lines compared to controls, it seems premature to conclude that H3K4me3 levels were increased at specific loci since the WT differentiation is a mixture of cell types at each time point compared to a seemingly more homogenous pool in the cJun mutant differentiations.

Response: We agree that the signal of H3K4me3 in bulk ChIP-seq data was averaged among the mixture cell types at each day. Although this is true, it led to the use of the KDMi, which indeed supports the role of H3K4me3 in CM differentiation.

4. In Figures 2G and 5C, the authors state that the cardiomyocytes have enhanced sarcomere structures. How was this determined or quantified? Likewise, in Movies EV3 and S4, how was beating strength measured or quantified?

Response: We counted the number of sarcomere structures as white boxes marked in each image related to Fig 2G and Fig 5E, and compared WT/control with KO/KDMi treatment (Box 4). From the bar plot, the number of sarcomere structures in both knock out c-JUN or treated by KDMi was more than WT or control group. Those results also add in Fig 2H and Fig 5F.

Beating strength was not measured, it was only a qualitative observation. Hence, we have changed the sentence to read: "Interestingly, we observed that spontaneous contractions of c-JUN KO cardiomyocytes

appeared to be increased, compared to the WT cells (Movies EV1 and EV2)” in the manuscript. (Lines 146-148)

5. From the single cell analysis, what were the distribution of the 13 clusters between WT and cJun mutant cells? It was reported in Figure 2 that almost all cells in the cJun mutant population at Day 15 were TNNT2+ cardiomyocytes, but it seems like in 6E many Day 15 mutant cells are epicardial, perhaps even more epicardial than WT differentiation (Figure 6D).

Response: We have split the t-SNE plot in WT and c-JUN KO to make the cell distribution clear (BOX5 A and B). At the same time, we have added this plot in Figure S3C.

We agree that in the FACS results 70%-90% cells were TNNT2+ in WT or c-JUN KO (Figure2), but in the single cell we got 57.8% and 24% TNNT2+ cells (including cardiomyocyte, smooth muscle, and CPMT, both of those cell types expressed TNNT2, BOX 5C) in D7 and D15 of c-JUN KO, 30% and 49% TNNT2+ cells (including cardiomyocyte, smooth muscle, and CPMT, both of those cell types expressed TNNT2, BOX 5C) in D7 and D15 of WT. As we show in BOX2, TNNT2+ cells are around 70% to 80% in the whole cell population. So the lower percentage of cardiomyocytes in both WT and KO may lead by the method used for cell clustering and cell type annotation. One another possible reason is we used a smaller cell filter (40µm) to filter cells before single cell RNA-seq library construction. This may drop out the big size cardiomyocytes.

6. Is there any explanation for why the KDM5 inhibitor would have the specific effect of activating cardiomyocyte genes and not other cell fates?

Response: We speculate that the KDM5i is working to 'boost' the already established differentiation direction that is determined by the cell culture medium. As the KDMi treatment could highly activate the marker genes of mesoderm and cardiac mesoderm cells such as TBXT/DKK1/EOMES, MEST/HAND2/KDR (Fig.1C and Fig. 5G-H). Hence, constant push the cells towards a CM cell fate.

Recently, a study published in *Exp Mol Med* suggested that inhibiting KDM5A could ameliorate pathological cardiac fibrosis ⁴. However, another study submitted to *BioRxiv* ⁵ suggests KDM5A could

regulate cardiomyocyte maturation by promoting fatty acid oxidation, oxidative phosphorylation, and myofibrillar organization. Those studies indicate a complex role of KDM5 in regulating cardiomyocyte cell fate decision and maturation. And we have added this speculation to the discussion. (marked in blue color; Lines 340-348)

7. Lines 224 and 306, the authors state that they utilized CHIP-seq to demonstrate c-Jun binding, but the methods suggest they used a different method, namely, CUT&Tag.

Response: We apologize for the mistake. c-JUN binding data was generated by CUT&Tag, and we have revised the manuscript to indicate this.

8. Why were clones #2 and #10 chosen over any of the other clones? No details are provided in the text.

Response: Picking and generating clones is a stressful process for the cells. In this case, clones #2 and #10 recovered first, and showed little evidence of spontaneous differentiation. Hence, we chose these two lines.

9. The manuscript and figures contained typos and grammatical errors that I would recommend fixing.

Below are a few that jumped out to me during the review process:

- a. Figure 1A, "Cardiac Mesoderm"
- b. Figure 4C and 4G, it should say ATAC-seq along the lefthand side
- c. Line 125, it should read, "we analyzed the time course RNA-seq data"
- d. Line 138-139, it should read, "c-Jun was expressed throughout hESC"
- e. Line 140, it should read, "and used CRISPR/Cas9 to delete"
- f. Line 168, "MESP1"
- g. Line 180, "chromatin accessibility is driven"
- h. Line 189, "we compared the"

Response: We thank the reviewer for pointing out these errors, which we have corrected. We have also carefully proof read the manuscript and have worked to improve the clarity and flow of the writing.

Reviewer #3: c-JUN has been previously identified as a critical TF for mouse embryonic development. In this manuscript, the authors discovered that c-JUN plays an important role in cardiomyocyte development using in vitro differentiation of human embryonic stem cells (hESCs). Loss of c-JUN leads to increased expression of RBBP5 and SED1B, thereby improving chromatin accessibility and deposition of H3K4me3 on regulatory elements associated with cardiomyocyte development. This manuscript is somewhat innovative, as c-JUN is known to be essential for normal cardiac development in mice but acts as a barrier in hESCs to cardiomyocyte transition.

Response: We thank the reviewer for their positive comments on our manuscript, and the points below which we have used to strengthen our work.

Major points:

1. Loss of c-JUN leads to early mouse embryonic death possibly due to failure to a normal cardiac system. develop. However, in this manuscript c-JUN is a barrier in hESCs to cardiomyocyte transition. There should be more explanation about the conflict function of c-JUN.

Response: Reviewer #1 (Point 4) also raised this point. Previous study shows that c-Jun defect leads to mouse embryonic lethal around E13.5 due to cardiac problems ¹. Here, our work shows c-JUN is a barrier to cardiomyocyte differentiation. However, it is important to highlight critical differences between the in vivo knock-out of c-Jun and the in vitro differentiation system employed here. Potentially, an accelerated generation of cardiomyocytes may be deleterious for proper heart generation due to reduced numbers of support cells. Indeed, our single-cell RNA-seq data indicates that the cardiomyocyte population expands at the expense of epicardial cells. Potentially, in the even more complex in vivo development of the heart, other cell types may also be defective, leading to a failure to establish the correct balance of cells required for a functioning heart organ. We have added the discussion about the contradictory phenomenon of c-JUN deletion between human and mouse early development in DISCUSSION section. (marked in blue color; Lines 349-359).

2. Are there any connections between the TFs predicted by SCENIC and H3K4me3? It would be valuable to supplement the discussion regarding the results shown in figure 6H.

Response: Thanks for the suggestion. When combined with the H3K4me3 data and SCENIC results, we can see H3K4me3 modification loci also enriched with EOMES/GATA/MEF TFs (Fig. 4D) that are predicted by SCENIC, indicating transcriptome, epigenetic modification, and chromatin dynamics were synergistic remodeling to reshape the cell fate to cardiomyocyte. We have discussed the connections between the TFs predicted by SCENIC and H3K4me3 in the revised manuscript. (marked in blue color; Lines 315-318)

Minor points:

1. The titles of video in the attachment do not align with the descriptions provided in the manuscript.

Response: We have corrected the title of the video.

2. There are issues with the titles of Result1, Result2 and Result6, as they are not complete sentences. Furthermore, it is necessary to further standardize the language used in other sections of the manuscript

Response: We have updated the manuscript and also invited a native English speaker to revise the manuscript.

3. At line 158, the sentence "rather than a facilitator" lacks supporting results or references in the text.

Response: We have deleted the sentence 'rather than a facilitator' to make the conclusion more accurate.

4. How can we determine that "the differentiation trajectory of hESCs to cardiomyocytes transition was similar between WT and c-JUN KO conditions" from figure 3H.

Response: We have revised the manuscript. As we show in the PCA plot, both WT and KO cells going to the similar direction. However, the KO cells appeared to be accelerated particularly D3-D5 and D7-D15 (BOX6). (Lines 209-211)

5. The "h" in figure4 should be capitalized as "H". Additionally, are there RNA-seq data available for RBBP5 and SED1B as supplementary material?

Response: We have corrected the error in Fig. 4. We have added RNA-seq data for RBBP5 and SETD1B as shown in the supplementary Figure S2 (BOX7). The raw data is available in the CSA submission: HRA003227.

6. There is some logical issue in lines 217-220. While chromatin accessibility can be regulated by histone methylation, the WB detection includes histone acetylation. This discrepancy should be addressed.

Response: Thanks for the comments, we have rewritten this sentence in the revised manuscript. (Lines 225-228)

7. The description in line 224 is not very clear.

Response: Thanks for the comments, we have rewritten this sentence in the revised manuscript. (Lines 228-232)

8. The font inconsistency in figure 5G should be addressed.

Response: Thanks for the comment, we have updated Figure 5 and kept all the fonts consistent.

9. The color scheme used for WT and KO in figure 6A cannot be clearly distinguished.

Response: We have updated Figure 6A-F and added one split WT and KO single cell t-SNE map in Fig. S3C to help illustrate the cell distribution.

10. The last sentence in line 278 has a weak correlation with the results presented in figure

Response: We agree this sentence is unclear. To clarify, we have rewritten the sentence 'After cell type annotation...'.(Lines 290-292)

References:

1. Eferl, R., Sibilio, M., Hilberg, F., Fuchsbichler, A., Kufferath, I., Guertl, B., Zenz, R., Wagner, E.F., and Zatloukal, K. (1999). Functions of c-Jun in liver and heart development. *The Journal of cell biology* *145*, 1049-1061. [10.1083/jcb.145.5.1049](https://doi.org/10.1083/jcb.145.5.1049).
2. Li, D., Liu, J., Yang, X., Zhou, C., Guo, J., Wu, C., Qin, Y., Guo, L., He, J., Yu, S., et al. (2017). Chromatin Accessibility Dynamics during iPSC Reprogramming. *Cell stem cell* *21*, 819-833 e816. [10.1016/j.stem.2017.10.012](https://doi.org/10.1016/j.stem.2017.10.012).
3. Zviran, A., Mor, N., Rais, Y., Gingold, H., Peles, S., Chomsky, E., Viukov, S., Buenrostro, J.D., Scognamiglio, R., Weinberger, L., et al. (2019). Deterministic Somatic Cell Reprogramming Involves Continuous Transcriptional Changes Governed by Myc and Epigenetic-Driven Modules. *Cell stem cell* *24*, 328-341 e329. [10.1016/j.stem.2018.11.014](https://doi.org/10.1016/j.stem.2018.11.014).
4. Wang, B., Tan, Y., Zhang, Y., Zhang, S., Duan, X., Jiang, Y., Li, T., Zhou, Q., Liu, X., and Zhan, Z. (2022). Loss of KDM5B ameliorates pathological cardiac fibrosis and dysfunction by epigenetically enhancing ATF3 expression. *Exp Mol Med* *54*, 2175-2187. [10.1038/s12276-022-00904-y](https://doi.org/10.1038/s12276-022-00904-y).
5. Deogharia, M., Agrawal, A., Shi, M., Jain, A.K., McHugh, K.J., Altamirano, F., Marian, A.J., and Gurha, P. (2023). Histone demethylase KDM5 regulates cardiomyocyte maturation by promoting fatty acid oxidation, oxidative phosphorylation, and myofibrillar organization. *bioRxiv*. [10.1101/2023.04.11.535169](https://doi.org/10.1101/2023.04.11.535169).

August 4, 2023

RE: Life Science Alliance Manuscript #LSA-2023-02121-TR

Prof. Dongwei Li
Guangzhou Medical University
Xinzao, Panyu District, Guangzhou, 511436, P.R.China
Guangzhou 511436
China

Dear Dr. Li,

Thank you for submitting your revised manuscript entitled "c-JUN is a barrier in hESCs to cardiomyocyte transition". We would be happy to publish your paper in Life Science Alliance pending final revisions necessary to meet our formatting guidelines.

- please address the final Reviewer 1's point regarding typos and grammar issues throughout the text
- please upload your main manuscript text as an editable doc file
- please upload your main and supplementary figures as single files
- please add a Running Title and a Summary Blurb/Alternate Abstract to our system
- please add ORCID ID for the corresponding (secondary and third corresponding) author--you should have received instructions on how to do so
- please add the Twitter handle of your host institute/organization as well as your own or/and one of the authors in our system
- please note that titles in the system and manuscript file must match
- please be sure to add all authors to the Author Contribution section in your manuscript file
- please upload your Tables in editable .doc or excel format;
- please add your main, supplementary figure, and table legends to the main manuscript text after the references section;
- please use the [10 author names et al.] format in your references (i.e., limit the author names to the first 10)
- if Figure S2 has two panels, add Panel B to the actual figure, its legend, and as a callout in the manuscript text. Otherwise, please delete the label of panel A in the figure, its legend, and in the callout
- please update your callouts for the Supplementary Movies in the manuscript Movie EV1A = Movie S1A) and also in legends in the manuscript file

A. FINAL FILES:

-- Summary blurb (enter in submission system): A short text summarizing in a single sentence the study (max. 200 characters including spaces). This text is used in conjunction with the titles of papers, hence should be informative and complementary to the title. It should describe the context and significance of the findings for a general readership; it should be written in the

present tense and refer to the work in the third person. Author names should not be mentioned.

B. MANUSCRIPT ORGANIZATION AND FORMATTING:

Sincerely,

Reviewer #1 (Comments to the Authors (Required)):

The authors have comprehensively addressed the majority of the raised questions, and now I can see the improvement of the paper. In general, I have no further issue regarding the acceptance of this manuscript. However, I still feel a bit concerned about the writing, as there are still some typo and Grammar issues. The author should carefully go through the whole manuscript and fix the mistakes.

Reviewer #2 (Comments to the Authors (Required)):

The manuscript from Zhong, Zhang, and Li et al. present work describing a role for c-Jun in cardiomyocyte differentiation repression. This updated manuscript greatly improves the authors conclusions and reinforces their conclusions over the original submission. I believe the authors have sufficiently addressed the comments of the other referees and myself.

Reviewer #3 (Comments to the Authors (Required)):

The authors completely addressed all my concerns. I think the paper is ready to publish.

August 7, 2023

RE: Life Science Alliance Manuscript #LSA-2023-02121-TRR

Prof. Dongwei Li
Guangzhou Medical University
Xinzao, Panyu District, Guangzhou, 511436, P.R.China
Guangzhou 511436
China

Dear Dr. Li,

Thank you for submitting your Research Article entitled "c-JUN is a barrier in hESC to cardiomyocyte transition". It is a pleasure to let you know that your manuscript is now accepted for publication in Life Science Alliance. Congratulations on this interesting work.

DISTRIBUTION OF MATERIALS:

Again, congratulations on a very nice paper. I hope you found the review process to be constructive and are pleased with how the manuscript was handled editorially. We look forward to future exciting submissions from your lab.

Sincerely,
